# Distinct ankyrin repeat subdomains control VAPYRIN locations and intracellular accommodation functions during arbuscular mycorrhizal symbiosis

Penelope L. Lindsay [1,2,3], Sergey Ivanov[1], Nathan Pumplin[1,2], Xinchun Zhang[1] & Maria J. Harrison [1] ✉

Over 70% of vascular flowering plants engage in endosymbiotic associations with arbuscular mycorrhizal (AM) fungi. VAPYRIN (VPY) is a plant protein that is required for intracellular accommodation of AM fungi but how it functions is still unclear. VPY has a large ankyrin repeat domain with potential for interactions with multiple proteins. Here we show that overexpression of the ankyrin repeat domain results in a *vpy*-like phenotype, consistent with the sequestration of interacting proteins. We identify distinct ankyrin repeats that are essential for intracellular accommodation of arbuscules and reveal that VPY functions in both the cytoplasm and nucleus. VPY interacts with two kinases, including DOES NOT MAKE INFECTIONS3 (DMI3), a nuclear-localized symbiosis signaling kinase. Overexpression of VPY in a symbiosis-attenuated genetic background results in a *dmi3*-like phenotype suggesting that VPY negatively influences DMI3 function. Overall, the data indicate a requirement for VPY in the nucleus and cytoplasm where it may coordinate signaling and cellular accommodation processes.

The mutualistic association of plants and arbuscular mycorrhizal (AM) fungi is widespread in terrestrial ecosystems and results in enhanced nutrition for both symbionts; the plant expands its access to soil mineral nutrients, particularly phosphorus[1] and the AM fungus gains an essential carbon supply (reviewed in ref. 2). The endosymbiotic association develops in the roots and involves intracellular growth of AM fungal hyphae through the epidermis and subsequently intracellular development of branched hyphae, known as arbuscules, in the cortical cells. In both cell types, the intracellular hyphae and the arbuscules are surrounded by a plant membrane and consequently the fungus is maintained in apoplastic compartments. Nutrient exchange between the symbionts occurs across the arbuscule-cortical cell interface[3–5].

During AM symbiosis, cellular remodeling occurs within the root cells to accommodate the fungus (reviewed in refs. 2,6). This includes cytoskeletal rearrangements, movement of the nucleus and secretory organelles to the vicinity of the growing hyphae and redirection of secretion to accomplish polarized membrane deposition around the arbuscule[7–11]. Transcriptional reprogramming, controlled at least in part by a symbiosis signaling pathway[12,13] regulates downstream gene expression and many of the cellular and metabolic changes. However, our current understanding of the pathway and its integration with other signaling pathways is incomplete. Additionally, signaling in the nucleus must be coordinated with the cellular rearrangements; yet the mechanisms of their integration are unclear.

VPY is a plant protein that is required for the intracellular phases of endosymbiosis with AM fungi and also with rhizobia. In *M. truncatula VPY* mutants, hyphal penetration of epidermal cells is reduced and while intercellular hyphae can develop in the cortex, arbuscule

[1]Boyce Thompson Institute, 533 Tower Rd., Ithaca, NY 14853, USA. [2]School of Integrative Plant Science, Plant Biology Section, Cornell University, Ithaca, NY, USA. [3]Present address: PLL: Cold Spring Harbor Laboratory, 1 Bungtown Rd, Cold Spring Harbor, NY 11724, USA. ✉e-mail: mjh78@cornell.edu

development does not occur[14–17]. Mutants lacking the *Petunia hybrida* ortholog of *VPY*, *PAM1*, are similarly impaired in arbuscule development with only an occasional occurrence of an intracellular hypha with a few small branches[16,18], and resistance to intracellular infection is associated with lignin-rich cell wall appositions[19]. In *M. truncatula*, *VPY* is also required for nodulation, specifically for the intracellular infection of rhizobia in the root hairs; *VPY* mutants show aberrant infection threads whose growth aborts in the root hair, and produce only a few small, nodule primordia[15,17,20]. These severe mutant phenotypes indicate that *VPY* is essential for endosymbiotic associations, yet its molecular function is unknown.

VPY is composed of an N-terminal VAMP-associated protein (VAP) / major sperm protein (MSP) domain coupled to an ankyrin repeat domain with nine ankyrin repeats[14–17]. While these domains exist broadly within eukaryotic proteins, this combination of the VAP/MSP and ankyrin repeat domains occurs only in plants and VPY is present only in AM symbiosis host plant species[21–23]. A related gene, VPY-like, exists both in AM symbiosis hosts and in non-host, *Physcomitrella patens*, where it influences development[24].

The VAP/MSP domain was first described in the major sperm protein of *C. elegans* sperm, a small protein that oligomerizes to enable sperm motility but that also functions as a signal[25,26]. In VPY, the predicted binding capacity of the VAP/MSP domain differs from that of the classical VAP/MSP; however in a yeast-two-hybrid assay, VPY was capable of interaction with a VAMP protein[27].

Ankyrin repeat domains function exclusively in protein-protein interactions and have the capacity to interact with diverse types of proteins. For example, Gankyrin, a protein with 7 ankyrin repeats, interacts with a cyclin-dependent kinase (CDK4), a ubiquitin E3 ligase, a proteasome subunit, a hepatocyte nuclear factor and a CCAAT/ enhancer binding protein[28–30]. The ability of ankyrin repeat domains to interact with multiple proteins can lead to the linkage or coordination of distinct pathways[30]. With 9 ankyrin repeats as well as a VAP/MSP domain, VPY has broad potential for protein-protein interactions and initial knowledge of its interacting partners has been obtained. VPY interacts with an EXOCYST complex member, EXO70I, which plays a role in exocytosis during development of the periarbuscular membrane[31]. EXO70I co-localizes with VPY in puncta adjacent to the periarbuscular membrane but the nature and significance of these punctate accumulations are unclear[31]. Similarly, VPY interacts with EXO70H4 during root hair infection by rhizobia, and also with a ubiquitin E3 ligase called LUMPY INFECTIONS (LIN)/CERBERUS[17,20]. These proteins co-localize in puncta ahead of growing infection threads[20] and LIN/ CERBERUS appears to stabilize VAPRYIN[17]. While these studies associate VPY with polarized exocytosis, VPY shows a broader nucleo-cytoplasmic location and the *upy* mycorrhizal phenotype[14–17] argues for additional functions beyond an interaction with EXO70I.

Here we evaluate the ankyrin repeat domain and its contribution to VPY functions, subcellular location and protein-protein interactions.

## Results

### Punctate VPY accumulations do not co-localize with known endosomal markers and require both the VAP and ANK repeat domains

Initial studies of VPY-fluorescent protein fusions in mycorrhizal roots reported that VPY signals were visible in the nucleus, the cytoplasm and in prominent static and sometimes mobile puncta, often associated with arbuscule branch tips[14–16]. Here, we extended these analyses to i) evaluate the subcellular location of VPY in cells containing arbuscules at different stages of development; ii) determine whether the punctate VPY signals co-localize with markers of known endomembrane compartments and iii) determine whether either of the major VPY domains drive its accumulation in the puncta. Using a VPY-cpVenus fusion expressed from its native promoter, we observed the expected nuclear, cytoplasmic and punctate signals in colonized

cortical cells (Fig. 1a, b). The nuclear signal was present in cells containing arbuscules at all stages of their life cycle including cells in which the arbuscule had entirely collapsed. The nuclear signal was also visible in non-colonized cells, adjacent to those containing arbuscules (Fig. 1a). Punctate VPY signals were visible initially in cells with developing arbuscules, abundant in cells with fully branched arbuscules but absent in cells with collapsed arbuscules (Fig. 1b, c). Occasionally the punctate VPY signals showed a distinct crescent shape (Fig. 1b) as observed previously for EXO70I[31] with which it is known to associate. Crescent-shaped VPY signals co-localized with the co-expressed membrane marker (Supplementary Fig. 1).

To further investigate the identity of the punctate VPY signals, we co-expressed VPY-GFP from its native promoter with markers of early endosome, late endosome, autophagosomes, Endoplasmic Reticulum, *trans*-Golgi network and Golgi bodies each tagged with mCherry[32,33]. In colonized root cells, VPY-GFP signals did not co-localize with any of these markers (Supplementary Fig. 1, 2) indicating that the punctate VPY signals do not correspond to these endosomal compartments.

To determine whether either the VAP/MSP domain or the ankyrin repeat domain (ANK) directed accumulation in the puncta, the individual domains were fused with cpVenus. Signals from a cpVenus-VAP/MSP fusion (amino acids 1–134) and from an ANK-cpVenus fusion (amino acids 135–541) were visible in the nucleus and cytoplasm, but the punctate signals were undetected for the VAP/MSP domain fusion (Fig. 1d) and rare in the case of the ANK domain fusion (Fig. 1e). The VAP/MSP fusion protein is small, approximately 40 kDa, and therefore may enter the nucleus by passive diffusion; however, at 70 kDa the ANK fusion is likely too large to enter the nucleus by passive diffusion. VPY does not have a strongly predicted nuclear localization signal, so it is unclear how nuclear localization of VPY is attained.

Overall, we conclude that VPY is located in the nucleus prior to, and throughout arbuscule development and also in PAM-associated unidentified puncta, which increase in number as arbuscules develop but are absent in cells with collapsed arbuscules. Both the VAP/MSP and ankyrin repeat domains are necessary for VPY to accumulate in the puncta. The latter conclusion is consistent with the findings of Liu et al.[20], but differs from Bapaume et al.[27], possibly a reflection of native versus heterologous expression systems used in the respective experiments.

### Ankyrin repeats 3-5 are essential to support initiation and development of arbuscules, while ankyrin repeats 7–9 are required to enable arbuscules to grow to full size

The conserved domains analysis program (NCBI)[34] predicted two binding interfaces in the VPY ankyrin repeat domain, spanning ankyrin repeats 3–5 and 6–8. Within these interfaces, repeats 4, 5, 7 and 8 are highly conserved across dicot and monocot host species, while repeat 6 is very poorly conserved (Supplementary Fig. 3). Considering repeat conservation and interface prediction we generated the following VPY deletion proteins: VPY lacking the last ankyrin repeat (VPY$_{ANKΔ9}$), VPY lacking the last three ankyrin repeats (VPY$_{ANKΔ7-9}$), VPY with removal of just the three central repeats (VPY$_{ANKΔ3-5}$) and VPY with a large truncation (VPY$_{ANKΔ4-9}$). The full-length VPY protein and all deletion proteins were fused at their C-termini with cpVenus. The constructs were transformed into *upy-4* and into wild-type roots for complementation and sub-cellular location analyses, respectively. In all cases, the constructs were expressed from the native *VPY* promoter. For complementation, we focused on the mycorrhizal cortical cell phenotype, specifically the restoration of arbuscules (Fig. 2a). Full-length *VPY-cpVenus* and both *VPY$_{ANKΔ9}$-cpVenus* and *VPY$_{ANKΔ7-9}$-cpVenus* restored arbuscule formation in *upy-4* (Table 1, Fig. 2b); however, arbuscules in *upy-4* expressing *VPY$_{ANKΔ9}$-cpVenus* and *VPY$_{ANKΔ7-9}$-cpVenus* truncated proteins did not achieve full size. The expression of *VPY$_{ANKΔ9}$-cpVenus* resulted in arbuscules that were marginally smaller than wild type (Fig. 2c, d) while *VPY$_{ANKΔ7-9}$-cpVenus* resulted in arbuscules that were 50% the size of

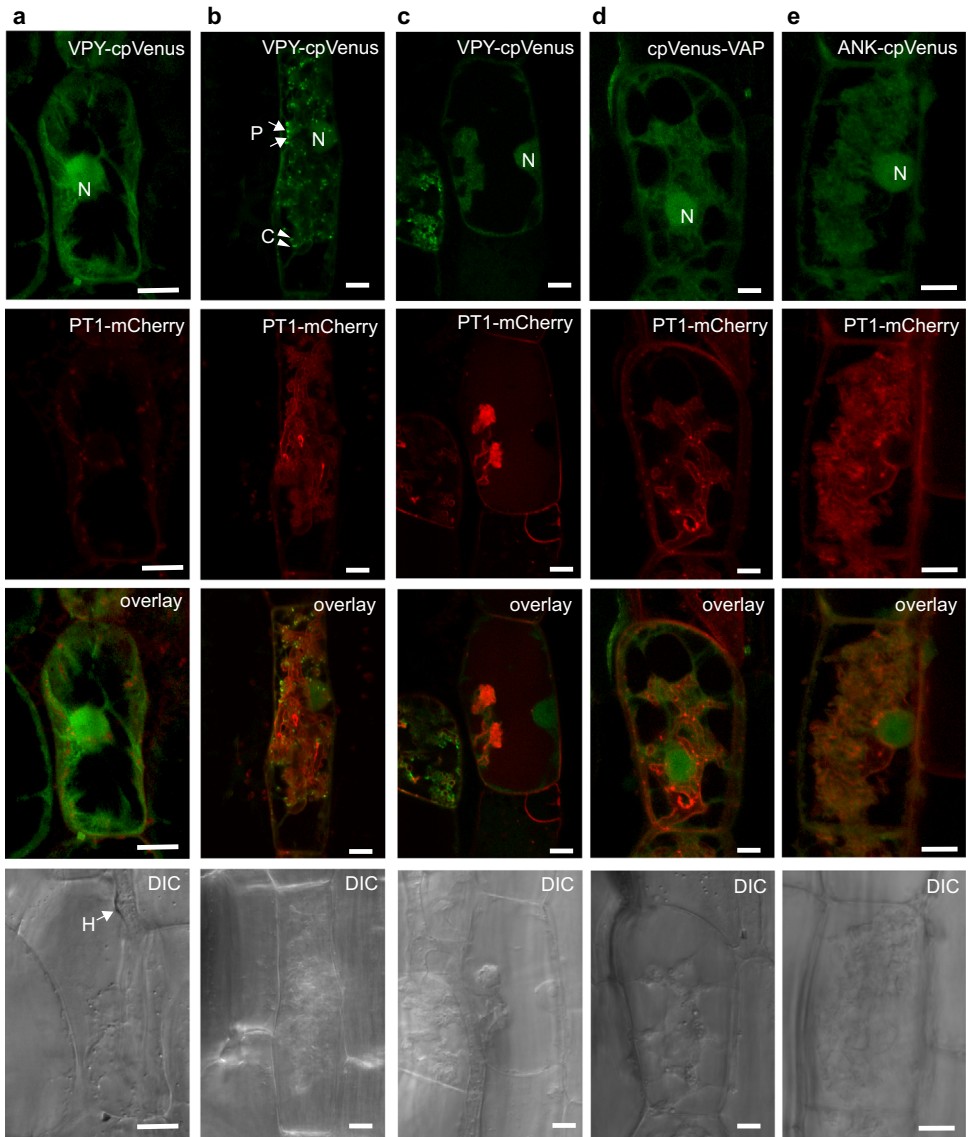

**Fig. 1 | Subcellular localization of full-length VPY, VAP/MSP, and ankyrin repeat (ANK) domains.** *Medicago truncatula* roots expressing either *VPYpro:VPY-cpVenus* (**a**–**c**)*, VPYpro:cpVenus-VAP* (**d**) or *VPYpro:ANK-cpVenus* (**e**) (green) and plasma membrane (PM) and periarbuscular membrane (PAM) marker *BCP1pro:PT1-mCherry* (red) colonized with *Rhizophagus irregularis*. Images of individual cortical cells prior to and during arbuscule development reveal the subcellular location of VPY (**a**) before AM fungal entry, (**b**) formation of fine branches, (**c**) fully collapsed arbuscule. Fluorescence images are projections of 10 optical sections on the *z*-axis taken at 0.5 μm intervals. Overlay images display both cpVenus and mCherry channels. Scale bars, 10 μm. H, hyphae; N, nucleus; P, puncta (arrow); C, crescent (arrowhead). DIC, differential interference contrast images of cells. Representative images of two independent transformations, with *n* = 25 arbuscule-containing cells observed in VPY-cpVenus, *n* = 43 arbuscule-containing cells in cpVenus-VAP, and *n* = 45 arbuscule-containing cells in ANK-cpVenus.

those in *vpy-4* transformed with *VPY-cpVenus* (Fig. 2c, e). In contrast, *VPYANKΔ3-5-cpVenus* and *VPYANKΔ4-9-cpVenus* did not restore arbuscule formation in *vpy-4* (Fig. 2b).

The subcellular location analyses revealed that VPYANKΔ9-cpVenus and VPYANKΔ7-9-cpVenus were present in the nucleus and cytoplasm but surprisingly, the abundant punctate signals were not observed for either of these truncated proteins (Fig. 2f). VPYANKΔ3-5-cpVenus was located only in the cytoplasm, with no punctate or nuclear signals (Fig. 2f). VPYANKΔ4-9-cpVenus was barely visible in the nucleus and did not show punctate signals, but accumulated in large cytoplasmic aggregations (Fig. 2f). Of these fusion proteins, only VPYANKΔ4-9-cpVenus is small enough to potentially enter the nucleus by diffusion.

In summary, the ankyrin repeat deletion proteins have differing capacities to restore arbuscule formation in *vpy-4*. VPY lacking ankyrin repeats 3–5 cannot support any arbuscule development. In contrast, VPY lacking repeats 7–9 supports arbuscule development but at a reduced level such that arbuscules do not attain full-size. The contribution from ankyrin repeat 9 is minor. Coupled with the functional data, the localization data suggest that arbuscule development does not require localization of VPY to the punctate location; however location in the puncta may be required to attain full-size arbuscules.

## Ankyrin repeat 5 is required for nuclear localization and VPYANKΔ5-cpVenus cannot complement *vpy-4*

Following the broad-scale ankyrin deletion analysis, we focused attention to individual ankyrin repeats within the two predicted binding interfaces and generated deletions of single ankyrin repeats 7 and 8, and then ankyrin repeats 4 and 5. Although ankyrin repeats 1 and 2 were not predicted as part of a binding interface, they are highly conserved in VPY proteins across plant species (Supplementary Fig. 3), so we generated deletion proteins lacking each of these ankyrin repeats also.

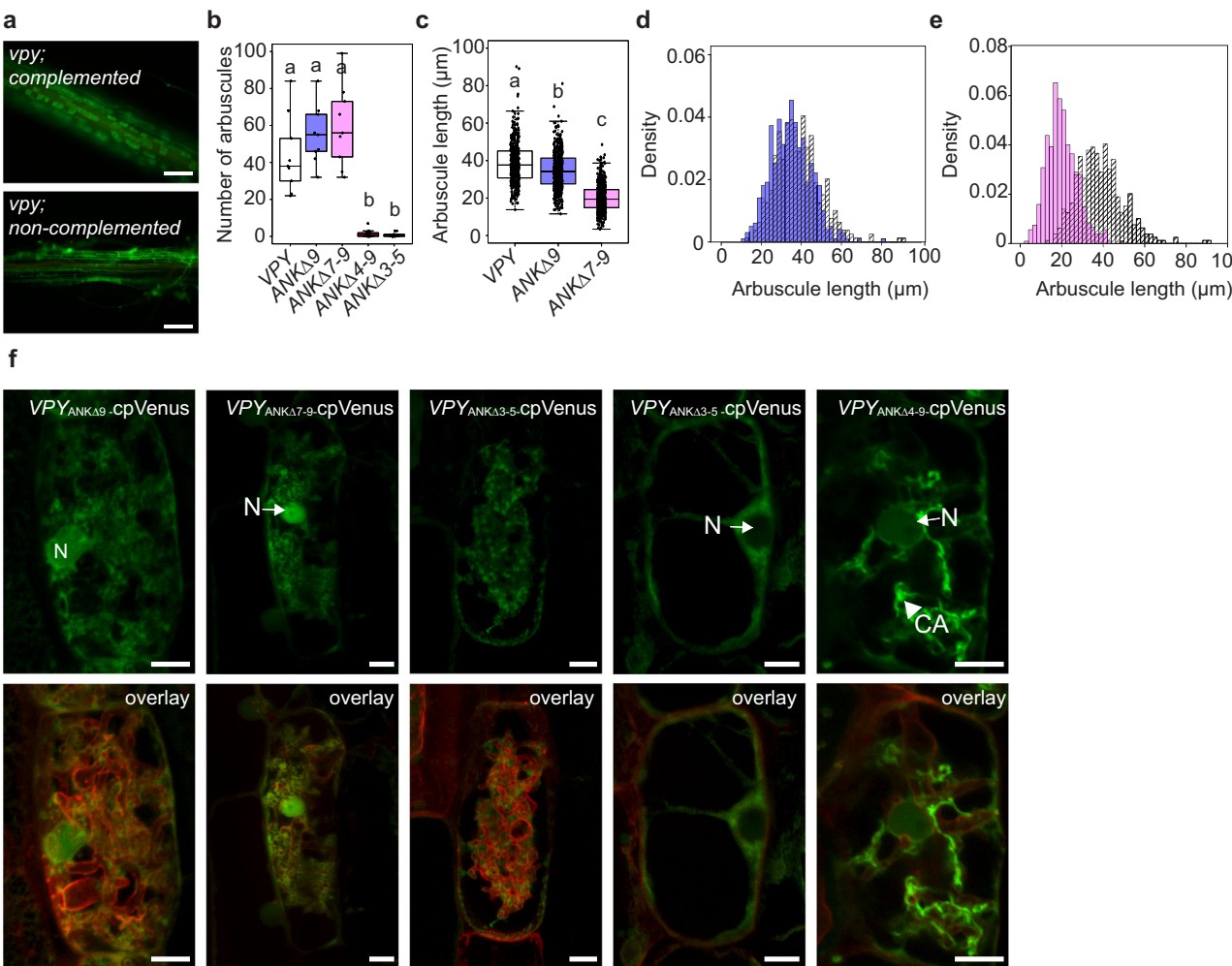

**Fig. 2 | Broadscale ankyrin deletion analysis. a–e** *vpy-4* was transformed with *VPY$_{pro}$:VPY-cpVenus* or ankyrin deletion constructs, colonized with *Diversispora epigeae*, and harvested 4 weeks post planting. Roots were stained with wheat germ agglutinin conjugated to AlexaFluor 488 (WGA-AlexaFluor 488) to visualize the fungus. **a** Representative epifluorescence images of *vpy-4* WGA-AlexaFluor 488 stained roots complemented with *VPY$_{pro}$:VPY-cpVenus*, or not complemented, as seen for *vpy-4* transformed with *VPY$_{pro}$:VPY$_{ANKΔ3-5}$-cpVenus*. Scale bars, 100 μm. **b** Number of arbuscules per infection unit in *vpy-4* transformed with ankyrin deletion constructs. Different letters indicate statistically significant differences using one-way ANOVA, $p < 0.05$, $n = 9$ infection units, sampled from three biological replicates with three infection units per biological replicate. **c** Arbuscule length of *vpy-4* transformed with *VPY$_{pro}$:VPY-cpVenus* ($n = 396$ arbuscules), *VPY$_{pro}$:VPY$_{ANKΔ9}$-cpVenus* ($n = 496$ arbuscules), or *VPY$_{pro}$:VPY$_{ANKΔ7-9}$-cpVenus* ($n = 536$ arbuscules). Different letters signify statistically significant differences from one-way ANOVA, sampled from three biological replicates with three infection units per replicate. In (**b**)–(**d**), lines in boxplots represent the median value, box limits represent the upper and lower quartiles, whiskers represent 1.5 times the interquartile range. **d** Overlapping histogram of arbuscule length between

*VPY$_{pro}$:VPY$_{ANKΔ9}$-cpVenus* (purple bars) and *VPY$_{pro}$:VPY-cpVenus* (striped bars). Arbuscule length is plotted on the x-axis, and probability densities are plotted on the y-axis to normalize sample size. **e** Overlapping histogram of arbuscule length between *VPY$_{pro}$:VPY$_{ANKΔ7-9}$-cpVenus* (pink bars) and *VPY$_{pro}$:VPY-cpVenus* (striped bars). Arbuscule length is plotted on the *x*-axis, and probability densities are plotted on the *y*-axis to normalize sample size. **f** Subcellular localization of VPY ankyrin deletions. WT roots expressing vectors containing *VPY$_{pro}$:VPY$_{ANKΔ9}$-cpVenus*, *VPY$_{pro}$:VPY$_{ANKΔ7-9}$-cpVenus*, *VPY$_{pro}$:VPY$_{ANKΔ3-5}$-cpVenus*, or *VPY$_{pro}$:VPY$_{ANKΔ4-9}$-cpVenus* and PM and PAM marker *BCPI$_{pro}$:PT1-mCherry* were colonized with *R. irregularis* to determine the subcellular distribution of the truncated fusion proteins in arbuscule-containing cells. Overlay images display both cpVenus and mCherry channels. Samples are representative images of at least two independent experiments, with three root systems and ten root pieces imaged per experiment, $n = 47$ (*VPY$_{pro}$:VPY$_{ANKΔ9}$-cpVenus*), $n = 19$ (*VPY$_{pro}$:VPY$_{ANKΔ7-9}$-cpVenus*), $n = 44$ (*VPY$_{pro}$:VPY$_{ANKΔ4-9}$-cpVenus*), or $n = 22$ (*VPY$_{pro}$:VPY$_{ANKΔ3-5}$-cpVenus*) arbuscule-containing cells. Fluorescence images are projections of 10 optical sections on the *z*-axis taken at 0.5 μm intervals. Scale bars, 10 μm. N, nucleus (arrow); CA, cytoplasmic aggregate (arrowhead).

VPY$_{ANKΔ1}$-cpVenus failed to support arbuscule development and showed a potential vacuolar location (Table 1 and Fig. 3a, b, f). In contrast, VPY$_{ANKΔ2}$-cpVenus restored arbuscules in *vpy-4* and showed wild-type subcellular localization (Table 1 and Fig. 3a, b, f). VPY$_{ANKΔ7}$-cpVenus and VPY$_{ANKΔ8}$-cpVenus both failed to restore arbuscule formation in *vpy-4*, which was unexpected given that VPY$_{ANKΔ7-9}$-cpVenus supports the development of small arbuscules (Table 1). However, loss of a single repeat can result in longer range conformational aberrations, which may impact other interactions[35]. Both fusion proteins were visible in the same sub-cellular locations as full-length VPY so their lack of function is unlikely the result of protein instability (Fig. 3f).

VPY$_{ANKΔ4}$-cpVenus enabled arbuscule formation in *vpy-4* (Table 1 and Fig. 3d) but arbuscules were 25% smaller than those in *vpy-4* transformed with full-length VPY (Fig. 3e). In contrast, VPY$_{ANKΔ5}$-cpVenus was unable to restore arbuscule formation in *vpy-4* (Table 1 and Fig. 3d). In wild-type colonized roots, VPY$_{ANKΔ4}$-cpVenus signals were comparable to those of wild-type VPY-cpVenus (Fig. 3f), while VPY$_{ANKΔ5}$-cpVenus showed the typical punctate and cytoplasmic signals but the nuclear signal was absent (Fig. 3f). The inability of VPY$_{ANKΔ5}$-cpVenus to complement arbuscule development in *vpy-4*, could suggest a role for VPY in the nucleus. Alternatively, the deletion may disrupt its cytoplasmic function.

**Table 1 | Functionality and subcellular location of VPY ankyrin deletion constructs**

| Construct | Complementation | | Subcellular location | | |
|---|---|---|---|---|---|
| | Yes/No | Arb + / Total[a] | Nucleus | Cytoplasm | Puncta |
| VPY | Yes | 29/39 | Yes | Yes | Yes |
| VPY$_{ANKΔ7-9}$ | Yes[b] | 4/18 | Yes | Yes | No |
| VPY$_{ANKΔ3-5}$ | No | 0/10 | No | Yes | No |
| VPY$_{ANKΔ4-9}$ | No | 0/5 | Yes | Yes | No |
| VPY$_{ANKΔ9}$ | Yes[b] | 6/11 | Yes | Yes | No |
| VPY$_{ANKΔ8}$ | No | 0/15 | Yes | Yes | Yes |
| VPY$_{ANKΔ7}$ | No | 0/17 | Yes | Yes | Yes |
| VPY$_{ANKΔ5}$ | No | 0/17 | No | Yes | Yes |
| VPY$_{ANKΔ4}$ | Yes[b] | 7/12 | Yes | Yes | Yes |
| VPY$_{ANKΔ2}$ | Yes | 8/20 | Yes | Yes | Yes |
| VPY$_{ANKΔ1}$ | No | 0/13 | Yes | Yes | No |

[a] Indicates the number of plants with infection units containing arbuscules over the total number of plants assayed.
[b] Arbuscule size is reduced relative to wild type.

## Nuclear and cytoplasmic pools of VPY are required for wild-type VPY function

To determine whether accumulation of VPY in the nucleus is important for function, we generated constructs to alter the balance of VPY in the nucleus and cytoplasm. With the inclusion of an NLS signal, we were able to direct the majority of VPY-cpVenus to the nucleus and VPY became undetectable in the cytoplasm (Supplementary Fig. 4). To prevent accumulation of VPY in the nucleus, we identified a nuclear exclusion motif (NES), which when fused to VPY, successfully prevented nuclear accumulation of VPY-cpVenus (Supplementary Fig. 4). The *NLS-VPY-cpVenus* and *VPY-NES-cpVenus* constructs were expressed from the *VPY* promoter, and then protein location and ability to complement *upy-4* was assessed in *M. truncatula* roots (Fig. 4a). As observed earlier, VPY-cpVenus complemented *upy-4* and restored arbuscule development in 100% of the transgenic plants evaluated (*n* = 18 independent root systems), while arbuscules were not observed in any plants expressing NLS-VPY-cpVenus (*n* = 18 independent root systems). However, NLS-VPY-cpVenus did impact colonization of *upy-4* and hyphal entry through the epidermis was reduced relative to *upy-4* plants transformed with a cpVenus control construct (Fig. 4c). Additionally, infection unit length in *upy-4* plants expressing *NLS-VPY-cpVenus* was reduced relative to the control (Fig. 4d). These data suggest that over-accumulation of VPY exclusively in the nucleus negatively impacts its function. By contrast, *VPY-NES-cpVenus* enabled some arbuscule development in *upy-4* but infections differed in appearance relative to those complemented with *VPY-cpVenus* (Fig. 4b). Quantification revealed that infection units in *upy-4* expressing *VPY-NES-cpVenus* showed a higher number of intracellular hyphae (IH) and a reduced number of arbuscules relative to those in *upy-4* expressing *VPY-cpVenus* (Fig. 4e–g). So while a cytoplasmic VPY is sufficient to allow arbuscule development in *upy-4*, the aberrant intraradical hyphae:arbuscule (IH:A) ratio points to a requirement for VPY in the nucleus. Coupled with the *NLS-VPY* complementation phenotype, these data support the hypothesis that VPY function is achieved though action in the nucleus and cytoplasm.

## Overexpressing the complete ankyrin repeat domain in wild-type plants reduces arbuscule abundance

Given the importance of the ankyrin repeat domain and the expectation that it facilitates protein-protein interactions, we hypothesized that if overexpressed in wild-type roots, the ankyrin repeat domain would compete with the endogenous VPY for interactors and therefore act as a dominant-negative inhibitor. To test this hypothesis, we transiently overexpressed the full-length *VPY*, the *VAP/MSP* domain and the *ANK* domain in wild-type roots and evaluated the mycorrhizal phenotypes. Overexpression of the *ANK* domain, but not *VAP/MSP* domain or full-length *VPY*, increased the occurrence of infections units with a vpy-like phenotype ie., infection units lacking arbuscules (Fig. 5a–c) thus supporting the hypothesis.

## A candidate-based approach for identifying additional proteins that interact with VPY

The experimental data argue that the ankyrin domain mediates protein-protein interactions, while the deletion analyses indicate different functionality within the repeat domain which could suggest interactions with multiple proteins. Interactions of VPY with EXO70 proteins and a putative E3-ligase, LIN / CERBERUS[17,20,31] have been identified but nuclear-localized interactors have not been reported. To evaluate additional interacting proteins, we took a targeted co-immunoprecipitation approach with selected candidate genes that showed high co-expression with *VPY* (Table S1). Although co-expression of two genes does not necessarily indicate interaction of their encoded proteins, interaction can only occur if the proteins exist in the same cell type. *EXO70I* is highly co-expressed with *VPY* providing rationale for this approach[31]. In addition to six candidate genes highly co-expressed with *VPY* (*KIN1, KIN2, CKL1, CKL2, IPD3* and *TUBB1*) we included nuclear-localized proteins DELLA1 and DMI3 because their respective mutants show mycorrhizal phenotypes that partially overlap with that of *upy*. *upy* has a cortical cell mycorrhizal phenotype very similar to that of *della1 della2* and *upy* shows a reduced epidermal penetration phenotype similar, although not as severe, as that of *dmi3*[36,37]. Additionally, as outlined below, we had initial evidence for a connection between VPY and DMI3.

In each case, VPY and the candidate interacting proteins were tagged and co-expressed in *Nicotiana benthamiana* leaves and protein interactions assessed by co-immunoprecipitation. The *N. benthamiana* system has proven valuable for identifying pairwise interactions among proteins of the common symbiosis signaling pathway[38,39]. Of the candidate proteins evaluated by co-immunoprecipitation assays, two kinases, KINASE2 and DMI3 interacted with VPY, while KINASE1, Cyclin-dependent Kinase-Like 1, Cyclin-dependent Kinase-Like 2 and Interacting Protein of DMI3 (IPD3) did not interact with VPY (Table S1).

## VPY interacts with DMI3, a Calcium/ Calmodulin-dependent Protein Kinase

DMI3 was included in the interaction analysis because of similarity between the *dmi3* and *upy* mycorrhizal epidermal phenotypes and because we had discovered unexpectedly that overexpression of *VPY* in *ipd3-2 ipd3l-2* roots resulted in a *dmi3*-like phenotype; the fungus developed large, multi-lobed hyphopodia that mostly failed to penetrate the root epidermis (Fig. 6a-c). This did not occur in WT, *nsp1 nsp2* or *ram1* roots overexpressing *VPY* (Supplementary Fig. 5) and therefore these data raised the question of a potential interaction between VPY and DMI3. In co-immunoprecipitation assays, VPY-HA co-immunoprecipitated with GFP-DMI3, but not with GFP-ΔDELLA1, the other nuclear-located protein evaluated in this study (Fig. 6d). Additionally, reciprocal immunoprecipitation of VPY-GFP resulted in the co-immunoprecipitation of DMI3-FLAG (Supplementary Fig. 6). To examine the VPY-DMI3 interaction further, we tested the interaction of GFP-DMI3 with two ankyrin repeat deletion proteins; VPY$_{ANKΔ3-5}$-HA co-immunoprecipitated with GFP-DMI3, while VPY$_{ANKΔ7-9}$-HA did not, suggesting that ankyrin repeats 7-9 are required for the interaction with DMI3 (Fig. 6e, Supplementary Fig. 5). Furthermore, Medtr8g056020 a protein with four ANK repeats and a VAP domain, therefore similar in domain composition to VPY, also co-immunoprecipitated with DMI3-GFP (Supplementary Fig. 5). While the overall identity of VPY and Medtr8g056020 is low (28% amino acid identity), ankyrin repeats 7 and 8 of VPY share 40.6 % amino acid

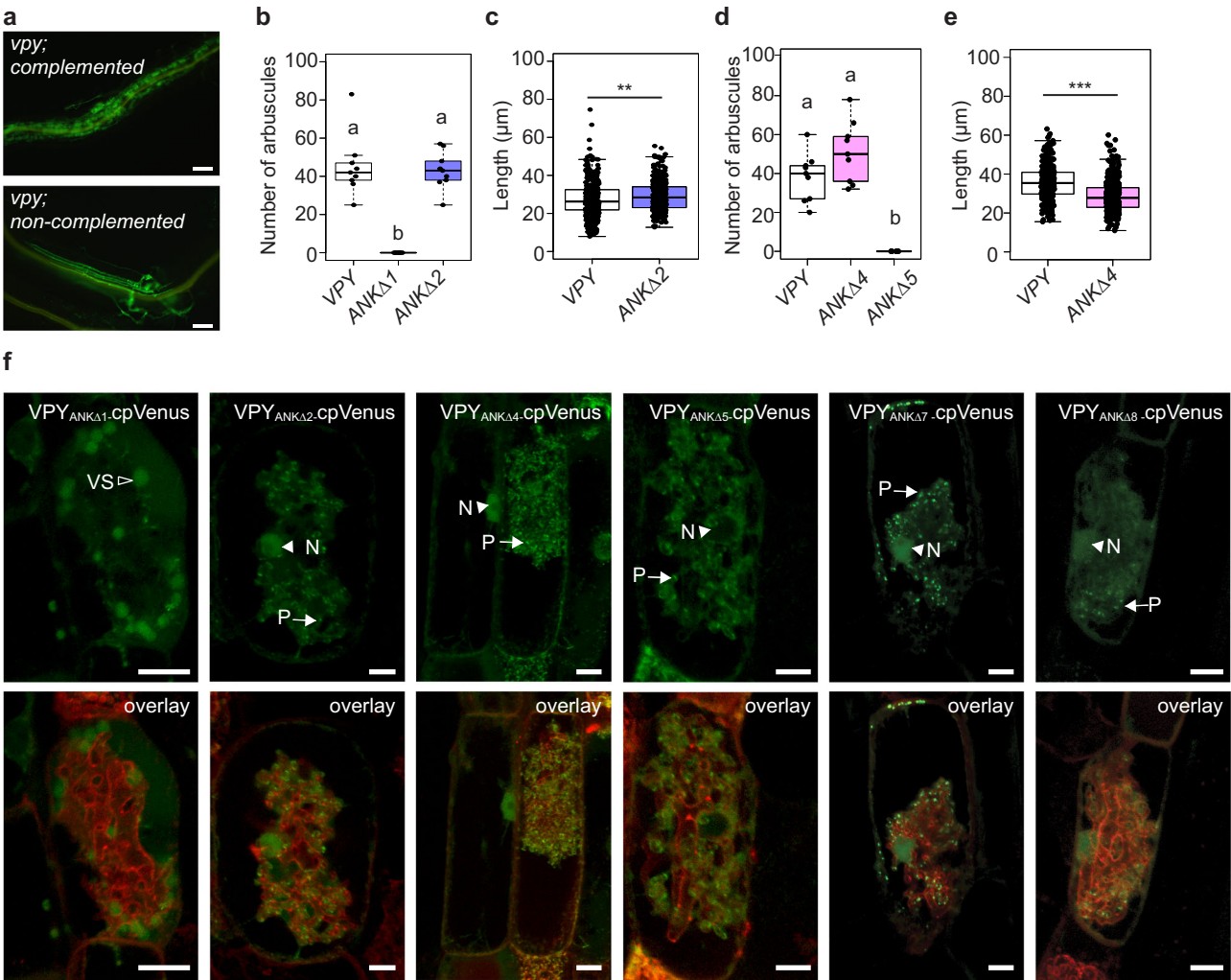

**Fig. 3 | Fine scale ankyrin deletion analysis. a–f** *vpy-4* was transformed with full-length *VPY_pro*:*VPY-cpVenus* or ankyrin deletion constructs, colonized with *D. epigeae*, and harvested 4 weeks post planting. **a** Representative epifluorescence images of *vpy-4* WGA-AlexaFluor 488 stained roots complemented with *VPY_pro*:*VPY-cpVenus*, or not complemented, as seen for *vpy-4* transformed with *VPY_pro*: *cpVenus*. Scale bars are 100 μm. **b** Number of arbuscules per infection unit in *vpy-4* transformed with ankyrin deletion constructs. Different letters indicate statistically significant differences using one-way ANOVA, $p < 0.05$, $n = 9$ infection units, sampled from three biological replicates with three infection units per biological replicate. **c** Arbuscule length of *vpy* transformed with *VPY_pro*:*VPY-cpVenus* ($n = 406$ arbuscules), or *VPY_pro*:*VPY_{ANKΔ2}-cpVenus* ($n = 388$ arbuscules), sampled from three biological replicates with three infection units per replicate. **d** Number of arbuscules per infection unit in *vpy-4* transformed with ankyrin deletion constructs. Different letters indicate statistically significant differences using one-way ANOVA, $p < 0.05$, $n = 9$, sampled from three biological replicates with three infection units per biological replicate. **e** Arbuscule length of *vpy-4* transformed with *VPY_pro*:*VPY-cpVenus* ($n = 344$ arbuscules), or *VPY_pro*:*VPY_{ANKΔ4}-cpVenus* ($n = 459$ arbuscules).

*** indicates a significance of $p < 0.001$ using a two-tailed Student's *t* test sampled from three biological replicates with three infection units per replicate. In (**b**)–(**e**), lines in boxplots represent the median value, box limits represent the upper and lower quartiles, whiskers represent 1.5 times the interquartile range. **f** WT roots were transformed with vectors containing *VPY_pro*:*VPY_{ANKΔ1}-cpVenus*, *VPY_pro*:*VPY_{ANKΔ2}-cpVenus*, *VPY_pro*:*VPY_{ANKΔ4}-cpVenus*, *VPY_pro*:*VPY_{ANKΔ5}-cpVenus*, *VPY_pro*:*VPY_{ANKΔ7}-cpVenus*, or *VPY_pro*:*VPY_{ANKΔ8}-cpVenus* and PM and PAM marker *BCP1_pro*:*PT1-mCherry* colonized with *R. irregularis* to determine the subcellular distribution of the truncated fusion proteins in arbuscule-containing cells. Samples are representative images of at least two independent experiments, with three root systems and ten root pieces imaged per experiment, $n = 37$ (*VPY_pro*:*VPY_{ANKΔ1}-cpVenus*), $n = 24$ (*VPY_pro*:*VPY_{ANKΔ2}-cpVenus*), $n = 33$ (*VPY_pro*:*VPY_{ANKΔ4}-cpVenus*), $n = 25$ (*VPY_pro*:*VPY_{ANKΔ5}-cpVenus*), $n = 38$ (*VPY_pro*:*VPY_{ANKΔ7}-cpVenus*), or $n = 14$ (*VPY_pro*:*VPY_{ANKΔ8}-cpVenus*) arbuscule-containing cells. Fluorescence images are projections of 10 optical sections on the z-axis taken at 0.5 μm intervals. Scale bars, 10 μm. N, nucleus (arrowhead), P, punctus (arrow), VS, vacuolar sphere (open arrowhead).

identity with ankyrin repeats 2 and 3 of Medtr8g056020 (Supplementary Fig. 7) including full conservation of three solvent-exposed residues which are not commonly found in ankyrin repeats (Supplementary Fig. 7) (Mosavi et al., 2004). Thus, although the interaction of DMI3 with Medtr8g056020 is unlikely to be biologically relevant, it may provide clues as to the residues within ankyrin repeats 7-9 that are significant for the DMI3/VPY interaction.

If interaction between VPY and DMI3 is required for the phenotype observed in the *ipd3-2 ipd3l-2* VPY overexpressing roots, then it would be predicted that this would not occur in *ipd3-2 ipd3l-2* overexpressing VPY_{ANKΔ7-9}. This is indeed the case, as overexpression of

the VPY_{ANKΔ7-9} deletion in *ipd3-2 ipd3l-2* did not result in the *dmi3*-like phenotype and the VPY_{ANKΔ7-9} overexpressing *ipd3-2 ipd3l-2* roots showed similar levels of epidermal penetration as *ipd3-2 ipd3l-2* roots overexpressing GFP (Supplementary Fig. 5). Taken together, the data suggest that VPY might negatively regulate DMI3 through a direct interaction.

## VPY interacts with a membrane-associated kinase, KINASE 2
KINASE2 (Medtr4g129010) is a serine-threonine kinase identified through a phylogenomics analysis as an AM symbiosis-conserved protein. A KINASE2 mutant (*kin2*) showed a significant reduction in

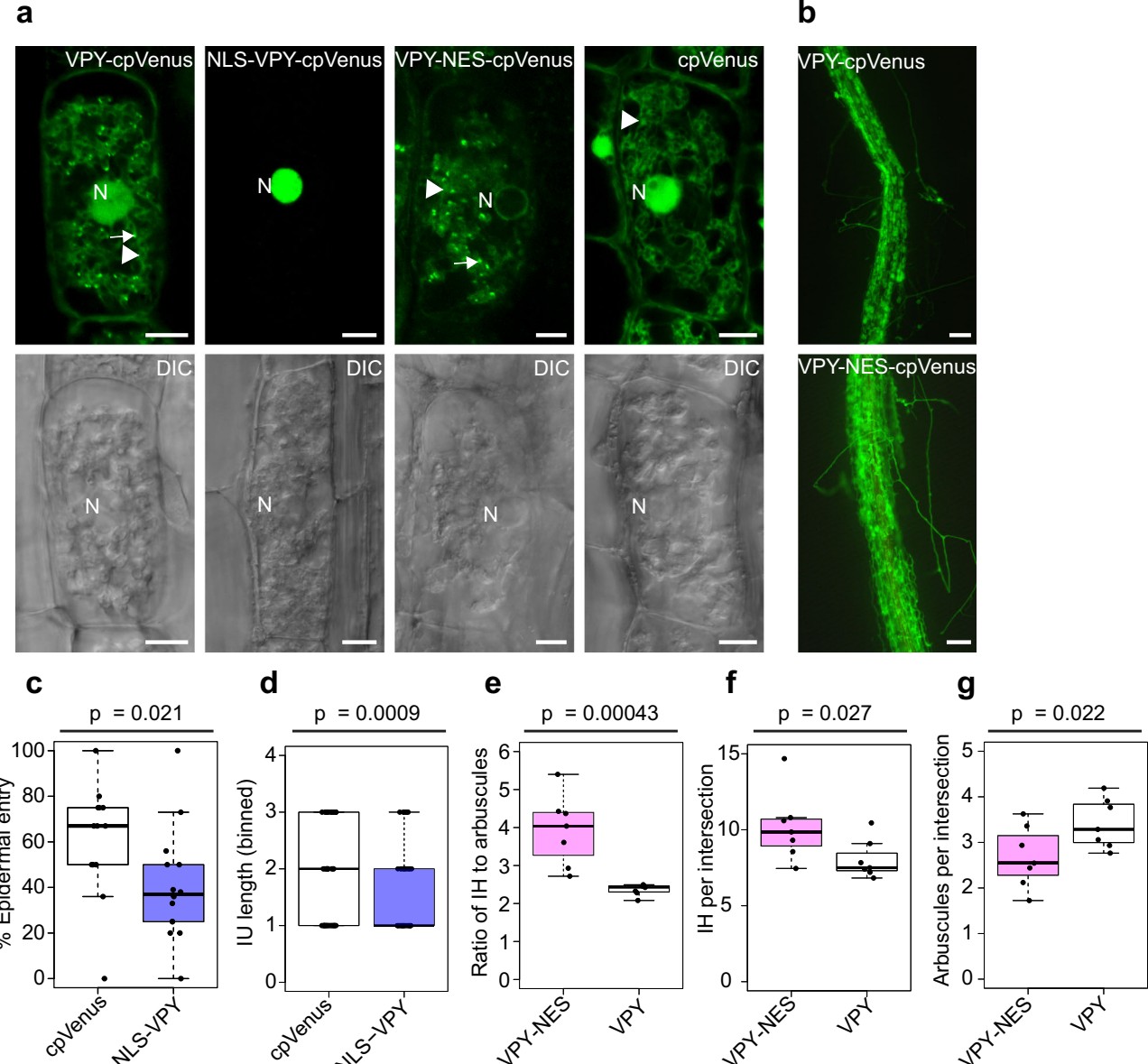

**Fig. 4 | Effect of NLS-VPY and VPY-NES on *upy-4*. a** Representative images of arbuscule-containing cells expressing *VPY_pro:VPY-cpVenus* (VPY, n = 28), *VPY_pro:NLS-VPY-cpVenus* (NLS-VPY, n = 16), *VPY_pro:VPY-NES1-cpVenus* (VPY-NES1, n = 19), or *VPY_pro:cpVenus* (cpVenus, n = 15). NLS-VPY accumulates exclusively in the nucleus, whereas VPY-NES1 is excluded from the nucleus. Scale bars are 10 μm. Arrows, puncta, arrowheads, cytoplasmic signal, N, nucleus. Three transgenic plants were imaged per construct. **b** Representative epifluorescence images of *upy-4* WGA-AlexaFluor 488 stained roots complemented with *VPY_pro:VPY-cpVenus* (VPY), or *VPY_pro:VPY-NES1-cpVenus* (VPY-NES). Scale bars are 100 μm. **c** The percentage of successful epidermal entry by *D. epigeae* in *upy-4* roots transformed with *VPY_pro:cpVenus* (cpVenus, n = 13) or *VPY_pro:NLS-VPY-cpVenus* (NLS-VPY, n = 14).

(**d**) binned infection unit (IU) length of *upy-4* roots transformed with *VPY_pro:cpVenus* (cpVenus, n = 42 infection units) or *VPY_pro:NLS-VPY-cpVenus* (NLS-VPY, n = 50 infection units) and colonized by *D. epigeae*, IUs from ≥ 13 root systems per construct. IU length was categorized into three bins, where 1 is 0–0.75 mm, 2 is 0.75–1.5 mm, and 3 is greater than 1.5 mm. **e** Ratio of intercellular hyphae (IH) to arbuscules, (**f**) IH per intersection, and (**g**) arbuscules per intersection in *D. epigeae*-colonized *upy-4* roots transformed with *VPY_pro:VPY-cpVenus* (VPY) or *VPY_pro:VPY-NES1-cpVenus* (VPY-NES1), n = 7 root systems and ≥202 intersections per construct. *P*-values were calculated with a one-tailed Student's *t* test. In (**c**)–(**g**), lines in boxplots represent the median value, box limits represent the upper and lower quartiles, whiskers represent 1.5 times the interquartile range.

overall colonization levels but did not affect fungal morphology[21]. Here we show that in mycorrhizal roots, *KINASE2* is expressed only in the colonized regions of the root with strong expression in arbuscule-containing cells (Fig. 7a). A KIN2−GFP fusion expressed from the native promoter localized at the plasma membrane and periarbuscular membrane and the protein was visible at all stages of arbuscule development and degeneration (Fig. 7b−e). KINASE2 has a myristoylation signature, which likely drives its localization to the plasma membrane.

Following expression in *N. benthamiana* leaves, immunoprecipitation of VPY-GFP resulted in the co-immunoprecipitation of KINASE2-HA (Fig. 7f), while GFP did not co-immunoprecipitate KINASE2-HA. It was noticeable that VPY-GFP co-immunoprecipitated KINASE2-HA efficiently, even when input VPY-GFP levels were low. By contrast, Co-immunoprecipitation suggests a slight interaction of Medtr8g056020-GFP with KINASE2, which is just visible when Medtr8g056020-GFP input is high. KINASE2 has a sister clade homolog, KINASE1, which lacks the myristoylation signature but shares 60 % amino acid identity with

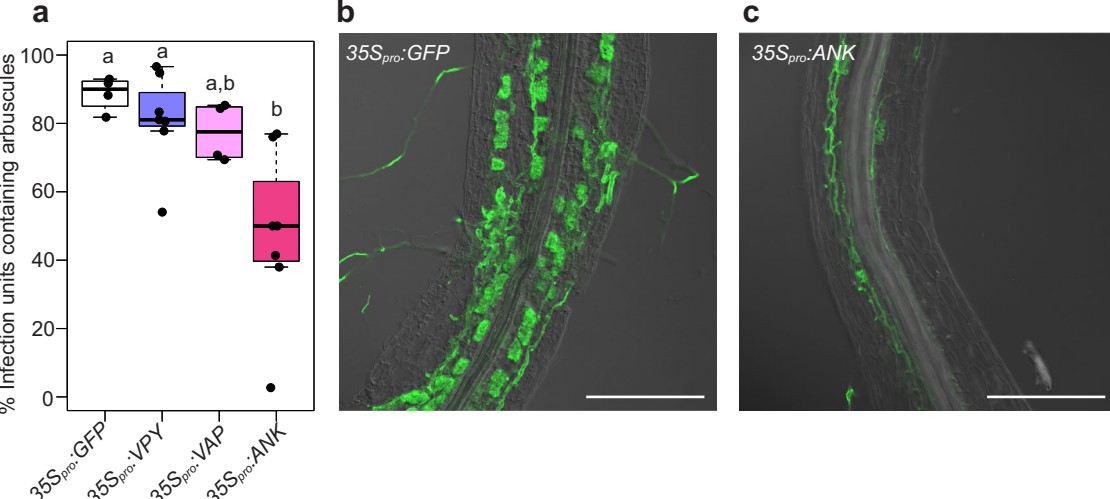

**Fig. 5 | Overexpressing the ankyrin repeat domain (ANK) in WT plants reduces arbuscule abundance.** WT roots were transformed with $35S_{pro}$:*VPY* ($n = 7$ root systems), $35S_{pro}$:*VAP* ($n = 4$ root systems), $35S_{pro}$:*ANK* ($n = 7$ root systems), or a negative control construct, $35S_{pro}$:*GFP* ($n = 4$ root systems) and colonized with *D. epigeae*. **a** Percentage of infection units containing arbuscules in transgenic roots. Different letters indicate statistically significant differences using one-way ANOVA, $p < 0.05$. Lines in boxplots represent the median value, box limits represent the upper and lower quartiles, whiskers represent 1.5 times the interquartile range. Confocal image of an infection unit in a WGA-AlexaFluor488 stained root expressing $35S_{pro}$:*GFP* (**b**) or $35S_{pro}$:*ANK* (**c**). Images show typical infection units in the GFP or ANK overexpressing roots. Scale bars, 250 μm. Images are overlays of fluorescence and DIC.

KINASE2 and shows a similar expression pattern in colonized roots (Supplementary Fig. 8). VPY did not co-immunoprecipitate KINASE1 (Supplementary Fig. 8), although we note that KINASE1 protein did not accumulate well in these assays. In an additional experiment with FLAG-tagged KINASE2, VPY-GFP was again able co-immunoprecipitate KINASE2, a further confirmation of their interaction (Supplementary Fig. 6).

Several VPY ankyrin repeats feature a TXXH domain (Supplementary Fig. 3), which in other ankyrin domain-containing proteins influences the stability of ankyrin repeats, and can be phosphorylated[40,41]. We evaluated the ability of KINASE2 to phosphorylate VPY in an in vitro phosphorylation assay. KINASE2, but not a catalytically inactive variant, KINASE2$_{K70N}$, was able to autophosphorylate and to trans-phosphorylate myelin basic protein but phosphorylation of VPY was not observed (Fig. 7g).

## Discussion

Essential for endosymbiosis and comprised of two well-characterized protein domains whose union in a single protein is reported only in plants, the molecular functions of VPY are still a mystery. VPY loss-of-function mutant phenotypes indicate that the protein is required for hyphal growth into cells, while VPY interactions with EXO70 proteins predict a role in the development of perimicrobial membranes. However, additional functions and additional interaction partners are likely, and the accumulation of VPY in the nuclei of cells prior to hyphal entry raises the question of a role in the nucleus, where pre-entry events, including signaling, transcriptional changes and DNA replication have been observed[42–44]. Here we extend current knowledge of VPY and specify ankyrin repeat binding interfaces required for subcellular localization and accommodation functions. Through forced accumulation or export from the nucleus, we provide evidence that both nuclear and cytoplasmic VPY pools are needed for wild-type function. Interactions with Kinase 2, a membrane-anchored kinase and DMI3, a nuclear-localized kinase, suggest that VPY may influence signaling during symbiosis, and this suggestion is further supported by phenotypic data.

We chose to focus on the ankyrin repeat domain as a starting point for structure-function and subcellular location analyses of VPY.

This large domain comprises more than 75% of the protein and its 9 ankyrin repeats offer considerable potential for interactions with other proteins. Previous structural analyses of proteins with extended ankyrin repeat domains have shown that three to five ankyrin repeats constitute a protein binding interface and these can function quasi-independently, so that disruption of one interaction space does not necessarily influence binding capabilities of another interface[35,45]. Consequently, deletion analyses of extended ankyrin repeat domains have proven informative as illustrated for Drosophila giant Ankyrins[46]. Here, deletion analyses identified binding interfaces formed by ankyrin repeats 3–5 and ankyrin repeats 7–9 as significant for VPY function. Our experiments were performed prior to the release of AlphaFold[47], but it is interesting to view AlphaFold structural predictions in the light of our experimental data. AlphaFold predicts a 'caliper'-like shape for VPY with ankyrin repeats 1–5 separated from repeats 7-9 by the less-ordered helix of ankyrin repeat 6 that forms a hinge-like structure between the two interfaces (Supplementary Fig. 9). Predictions of the ankyrin repeat deletion proteins suggest that removal of single repeats does not alter the overall structure substantially, while removal of repeats 7-9 results in loss of a discrete section and flexible C –terminal tail, but leaves the larger repeat interface and the VAP domain intact. Our experimental data showed that VPY$_{ANKΔ7−9}$ enables arbuscule development of limited size, which supports the functional significance of the larger ankyrin repeat region and VAP domain for allowing hyphal entry into cells and initiation of arbuscule development. AlphaFold predictions of the C-terminal tail and possible fold back onto the ankyrin repeats, are intriguing. Future analyses of this region are warranted, particularly in light of similar autoregulatory mechanisms in the ankyrin domain protein myosin 16[48].

A feature that has intrigued VPY researchers is its accumulation in punctate locations within the cell[14,16,20]. Previous studies have suggested that VPY puncta may be the predominant site for its potential exocytosis-related function[20,31] however, our current analyses suggest that that accumulation of VPY in puncta is not a prerequisite for enabling arbuscule development. The VPY$_{ANKΔ9}$ protein restores arbuscule development in *vpy-4* and arbuscules attain almost full size, but the VPY$_{ANKΔ9}$ protein does not accumulate in visible puncta. It is possible that VAPRYIN puncta are present in arbuscule-containing cells

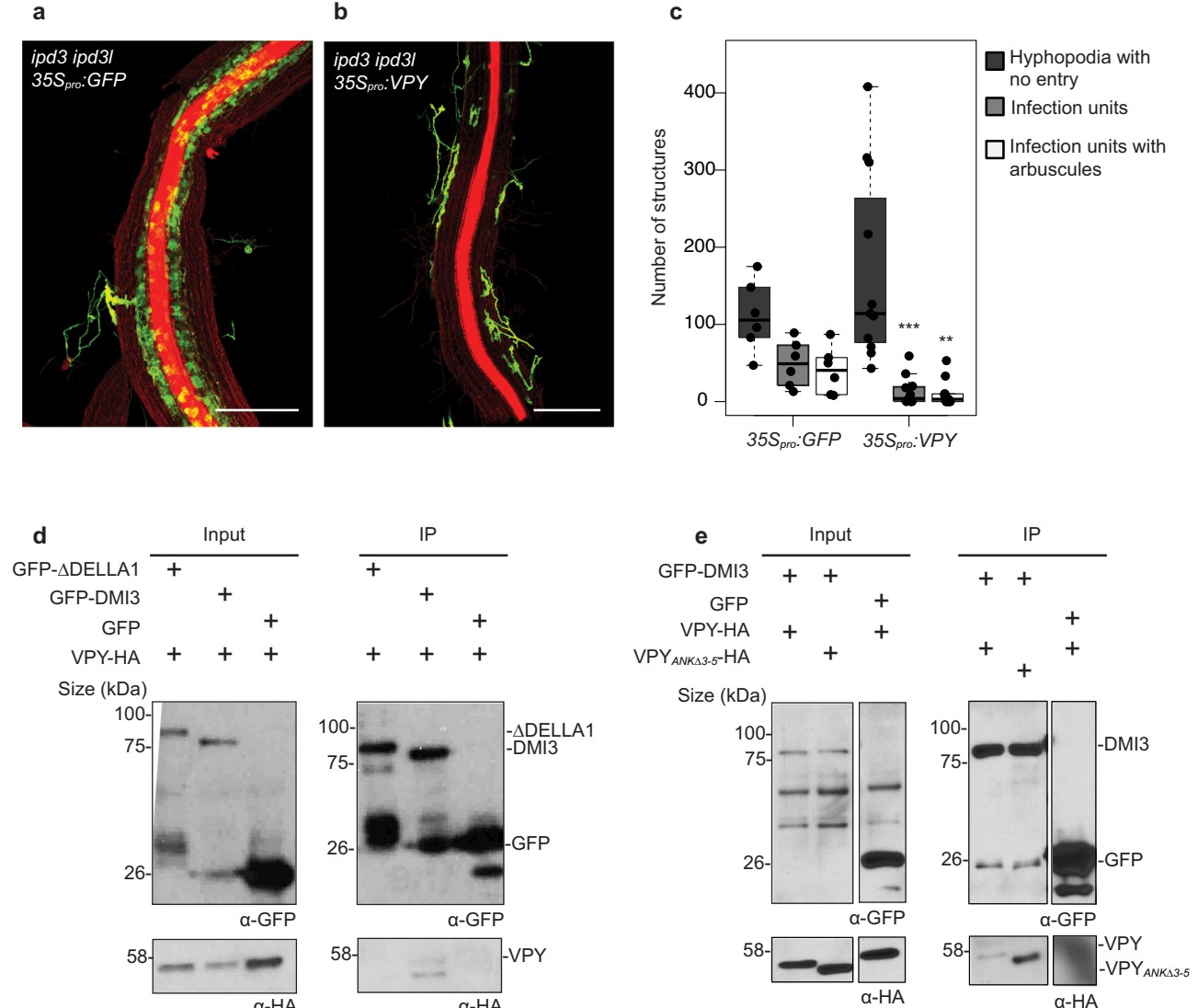

**Fig. 6 | Overexpression of VPY in *ipd3 ipd3l* results in a *dmi3*-like phenotype and VPY interacts with DMI3. a–c** *ipd3-2 ipd3l-2* roots transformed with *35Spro:VPY* or a negative control, *35Spro:GFP*, colonized with *D. epigeae*, and harvested three weeks post inoculation. Roots were stained with WGA-AlexaFluor488 (green) and propidium iodide (red) to reveal the fungus and plant cell walls, respectively. Roots overexpressing VPY show hyphopodia that fail to enter the epidermis. Scale bar, 500 μm. **c** Quantification of surface hyphopodia, infection units, and infection units containing arbuscules in *ipd3-2 ipd3l-2* roots overexpressing *35Spro:GFP* ($n = 6$ root systems) and *35Spro:VPY* ($n = 11$ root systems). ** indicates $p < 0.01$, *** indicates $p < 0.001$ using a two-tailed Student's $t$ test, *35Spro:VPY* as compared to *35Spro:GFP*. Lines in boxplots represent the median value, box limits represent the upper and lower quartiles, whiskers represent 1.5 times the interquartile range. **d** VPY and DMI3 interact in a co-immunoprecipitation analysis in *N. benthamiana*. *VPY-HA* and either *GFP-DMI3* or *GFP-ΔDELLA1* or *GFP* were co-expressed in *N. benthamiana* leaves using *35Spro* and GFP-tagged proteins were immunoprecipitated using anti-GFP magnetic trap beads. The presence of VPY-HA was assessed via western blot. VPY-HA co-immunoprecipitated with GFP-DMI3 but not with GFP-ΔDELLA1 or GFP. **e** *VPY-HA* and *VPYANK*<sub>Δ3-5</sub>-HA were co-expressed with *GFP-DMI3* or *GFP* in *N. benthamiana* leaves under the *35Spro* and GFP-tagged proteins were immunoprecipitated using anti-GFP magnetic trap beads. The presence of VPY-HA and VPYANK<sub>Δ3-5</sub>-HA was assessed by western blot. In both (**d**) and (**e**), co-immunoprecipitations were repeated once with similar results.

expressing VPY<sub>ANKΔ9</sub> but at levels below our detection. If so, our data still suggest that the massive accumulation of VPY in puncta is not required to initiate arbuscule development. In wild-type roots, VPY puncta accumulate gradually in the cortical cells over the course of arbuscule growth, with many puncta visible when the arbuscule is fully branched. Perhaps the punctate accumulations fulfill a late growth regulatory function rather than a 'reservoir function' to enable rapid deployment to the periarbuscular membrane as originally proposed[31]. It is also possible that the puncta are a consequence of high VPY protein levels, which might also explain why they are visible when VPY is overexpressed in heterologous systems[27].

In addition to cytosolic and punctate locations, VPY-GFP signals are occasionally visible in crescent-shaped regions at the periarbuscular membrane around arbuscule branch tips, particularly in young arbuscules (Fig. 1, Supplementary Fig. 1). Similar crescent-shaped signals were reported previously for components of the exocyst[49,31] and it is likely that these indicate a transient location of VPY and EXO70I[31] at the periarbuscular membrane. The interaction of VPY with KINASE2 could potentially occur at this location as KINASE2 is located at the periarbuscular membrane throughout arbuscule development. KINASE2 did not phosphorylate VPY in in vitro assays; however, it is possible that VPY modulates kinase activity thereby altering phosphorylation of KINASE2 targets as observed for other ankyrin repeat proteins[50–53]. Given the location at the periarbuscular membrane, future investigation of EXO70I as a kinase target is warranted as EXO70s can be regulated by phosphorylation[54].

*VPY* expression is induced rapidly in response to LCO signaling[55] and VPY accumulates in the nucleus of epidermal cells and cortical

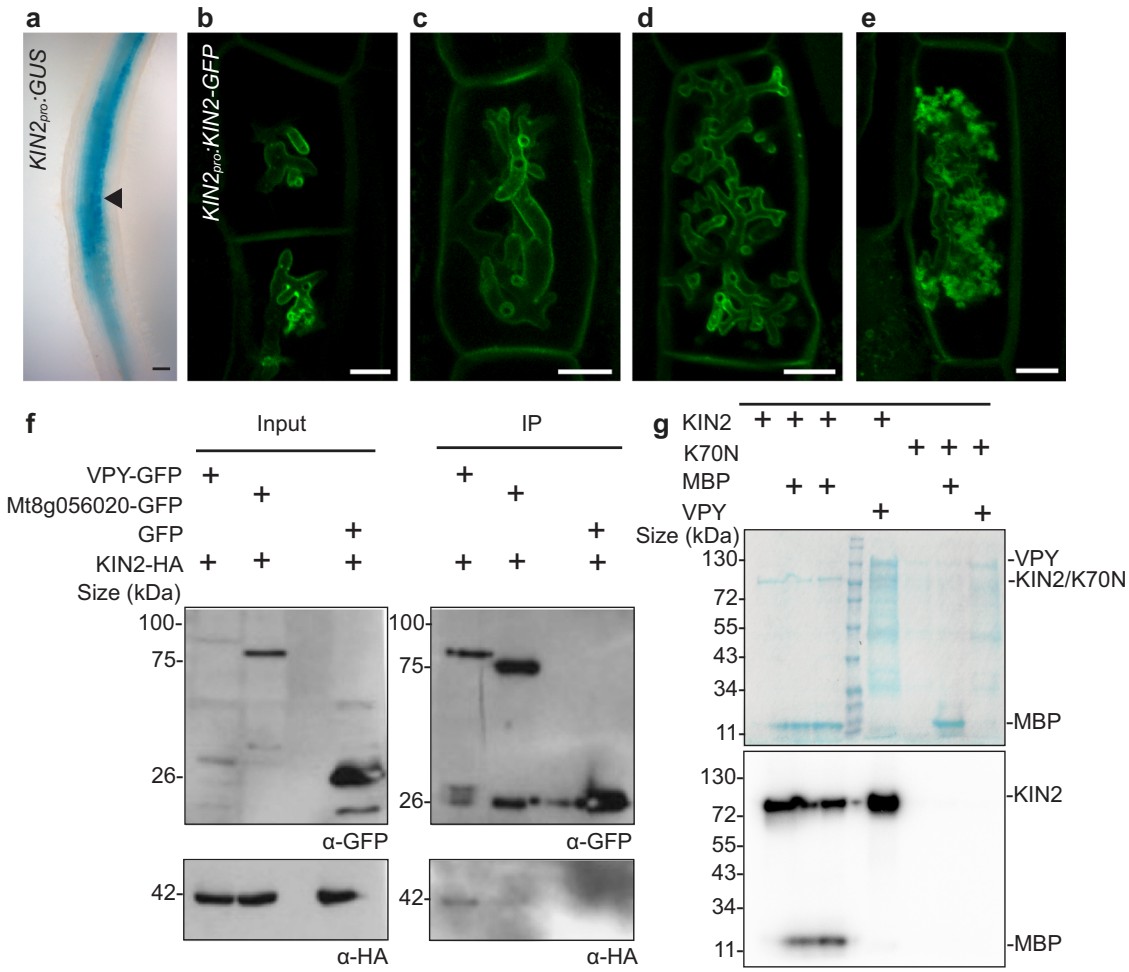

**Fig. 7 | VPY interacts with the PAM-localized kinase, KINASE2. a** Colonized root expressing *KIN2pro:GUS*. Scale bar, 100 μm. Representative image, *n* = 5 transgenic root systems and 15 infection units. **b**–**e** Cells containing arbuscules from roots expressing *KIN2pro:KIN2-GFP*. KIN2-GFP locates at the periarbuscular membrane and plasma membranes in cells with arbuscules at all stages of development and degeneration. Scale bars are 10 microns. Representative image of *n* = 60 arbuscule-containing cells. **f** Co-immunoprecipitation assay with VPY and KIN2. VPY-GFP, or controls, Medtr8g056020-GFP (Mt8g056020-GFP) or GFP were co-expressed in *N.* *benthamiana* with KIN2-HA. Proteins were immunoprecipitated using GFP trap magnetic agarose beads and the presence of KIN2-HA was assayed via western blot. This experiment was repeated once with similar results. **g** in vitro phosphorylation assay with KIN2. KIN2, or its catalytically inactive variant, KIN2_{K70N} were expressed in *E.coli*. Partially purified proteins were mixed with [32 P]γ -ATP, and auto-phosphorylation and trans-phosphorylation of myelin basic protein (MBP) or VPY was assessed. KIN2 shows kinase activity but did not phosphorylate VPY. This experiment was repeated five times with similar results.

cells prior to hyphal entry[14] (Fig. 1a). Location in the nucleus suggests a role perhaps distinct from the cytoplasmic post cell-entry roles[20,31] that have formed the focus of most studies[20,31]. Here, deletion of ankyrin repeat 5 resulted in a VPY mutant protein that showed typical accumulation in the cytoplasm and puncta but not in the nucleus, and failed to complement *vpy-4*. This might hint at a nuclear role but the loss of ANK5 could also result in a defective protein. To further evaluate a requirement for VPY in the nucleus, we forced accumulation in the nucleus and also prevented accumulation in the nucleus with the addition of NLS or NES motifs, respectively. NLS-VPY exacerbated the *vpy-4* phenotype, indicating that increased VPY in the nucleus has a negative influence on hyphal penetration into epidermal cells and on lateral spread in the cortex. A negative influence on hyphal entry into cortical cells could not be evaluated because hyphal entry into cortex cells is almost fully blocked in *vpy-4*. Although expressed from the *VPY* promoter, it is possible that the NLS-VPY construct essentially results in VPY overexpression in the nucleus as the entire VPY cytoplasmic pool is driven to the nucleus. Thus the effects observed could arise from a nuclear overexpression phenotype and possibly a dominant-negative effect. In contrast, VPY-NES supports intracellular growth in the cortical cells and arbuscule development but with an altered pattern of development. Intercellular hyphal development increases while arbuscule numbers decrease. Visually, the infection units have an altered appearance and retain many short intercellular hyphal branches that are typical of *vpy* mutants but not of wild type. The *vpy-4* phenotype itself, shows an altered pattern of intercellular hyphal development with many cross branches. The VPY-NES phenotype may arise from complementation of aspects of intracellular accommodation without complementation of a nuclear function.

Through co-immunoprecipitation, we showed that VPY interacts with the nuclear-localized protein DMI3, an essential regulator of symbiotic signaling, which provides one possible avenue through which VPY might regulate hyphal entry into cells. DMI3 is required for transducing lipochitooligosaccharide/chitooligosaccharide signaling[56,57] which occurs both in the epidermis and cortex[42], although the *dmi3* mutant phenotype manifests in the epidermis at the first site of intracellular hyphal growth[37,58,59]. While a relationship between DMI3 and VPY has not been reported, overexpressing *VPY* in the mutant background lacking the interactor of DMI3, IPD3 IPD3L, prevented AM hyphal growth through the epidermis, which suggests there may be a negative regulatory relationship between VPY and DMI3. Furthermore, the phenotype observed in *vpy-4* expressing NLS-VPY could arise from VPY

sequestration of DMI3. Previous studies of CCaMK, the *Lotus japonicus* ortholog of DMI3, showed that negative regulation of CCaMK is required for hyphal entry into epidermal and cortical cells[60] so it is possible that VPY fine-tunes DMI3 activity, potentially in both cell types. VPY is one of the first genes, along with the symbiotic transcriptional regulator NSP1, to be induced following lipochitooligosaccharide treatment in a DMI3-dependent manner[55]. Furthermore, its expression in the epidermis of mycorrhizal roots is transient and no longer present once the fungus reaches the cortex[14]; so transient expression of *VPY* might also function to dampen DMI3 and reduce signaling in the epidermis once the fungus has entered the cell. There are several examples where the direct interaction of ankyrin repeat proteins and kinases negatively regulates kinase activity[50–53], so perhaps an analogous process occurs with VPY and DMI3. Currently, it remains to be determined whether the VPY interaction with DMI3 is important for hyphal penetration of cortical cells and/or whether VAPRYIN has additional partners in the cortical cell nuclei that regulate this step of the mycorrhizal association. Based on the interaction data, the ankyrin repeat 7-9 interface was significant for interaction with DMI3, but the VPY$_{ANKΔ7-9}$ protein complemented arbuscule development in *vpy-4* with arbuscules showing a defect in size, which suggests that additional VPY interactors contribute to its role in enabling hyphal entry into cortical cells. Nevertheless, *DMI3* is expressed in cortical cells and we speculate that it could contribute to accommodation processes that enable arbuscule growth.

In summary, function and subcellular location analyses of VPY mutant proteins lacking one or more ankyrin repeat domains reveal a role for VPY in the nucleus, interactions with a membrane-located kinase and a nuclear-located kinase protein and provides a reason to reconsider the roles of VPY puncta. Based on these data and earlier studies, we propose that the strong *vpy-4* loss-of-function phenotype arises through the combined disruption of several processes necessary for symbiosis. This is true of other proteins with large ankyrin repeat domains, where complex phenotypes result from the simultaneous mis-regulation of multiple pathways[45,61,62] or the breakdown of a central hub[63]. Future work should clarify whether the *VPY* mutant phenotype arises from the loss of a scaffolding function that connects disparate proteins with one another, or if VPY modulates protein activity through binding to its interactors individually.

## Methods

### Identification of *vpy-4* mutant

The *vpy-4* allele (A17 background) was identified via Targeting Induced Local Lesions in Genomes (TILLING) conducted by *RevGen*UK, John Innes Center Genome Laboratory, Norwich, UK. Line 5162 had a C to T transition at nucleotide 433, which results in a premature stop codon at amino acid 145, producing a truncated protein of 144 amino acids (Supplementary Fig. 10). Line 5162 was backcrossed twice and subsequently named *vpy-4*.

### Plant growth and inoculation with AM fungi

*Medicago truncatula* plants were inoculated and grown as described in[64]. Briefly, to characterize *vpy-4*, plants were grown in substrate containing 200 *Diversispora epigeae* spores and harvested three weeks post planting. For *VPY* overexpression and VPY ankyrin deletion experiments, plants were grown in substrate containing 200 *Diversispora epigeae* spores and harvested four weeks post planting. For the promoter GUS analysis, plants were grown in substrate containing 200 *Diversispora epigeae* spores or a mock-inoculum lacking spores and harvested three weeks post planting. For live imaging, *Medicago truncatula* roots were assayed four weeks post planting into growth medium containing 500-1000 *Rhizophagus irregularis* spores. For NES-VPY and NLS-VPY experiments, plants received 100 *Diversispora epigeae* spores and roots were harvested at 5 weeks post planting.

### *M. truncatula* transformation methods

*Agrobacterium rhizogenes*-mediated transformation of *M. truncatula* roots was conducted as described previously[64,65]. Briefly, lawns of *A. rhizogenes* strains containing plasmids of interest were grown on TY plates for 2 days at 28 °C. Sterilized A17 seeds were plated on petri dishes and inverted and incubated at 4 °C for 24 h in the dark to synchronize germination. Following cold treatment, seeds were incubated at 28 °C in the dark for 18 h to allow radicle growth. Seedlings with 1 cm radicles were selected for transformation. Seed coats were removed from seedlings, and the root tip was cut 3 mm from the tip. Cut radicles were dipped in the *A. rhizogenes* lawn and transferred to square plates containing modified F media lacking antibiotics. Plates were sealed with Parafilm and placed at a slant in an 18 °C incubator (16 h light/8 h dark) with a light intensity of 6.5 μm/m²s for 5 days. Then, plates were transferred to a growth chamber with a 16 h light (25 °C)/8 h dark (22 °C) regime with a light intensity of 50 μm/m²s for ten days. Transgenic roots were selected by screening for the presence of dsRed or GFP, depending on the construct used. Non-transgenic roots were excised from the plant, and plants were transferred to pots containing turface to let the transgenic root system develop further. After 10 days, transgenic roots were again selected, and were inoculated with AM fungi as described in the previous section.

*Agrobacterium tumefaciens*-mediated transformation of *M. truncatula* R108 with *VPY$_{pro}$:VPY-GFP* was carried out via transformation of leaf explants and regeneration via somatic embryogenesis following the detailed step-by-step protocols as reported[66–68].

### WGA-Alexafluor 488 staining and visualization of AM fungi

To visualize AM fungi within roots, roots were placed into six well plates with 50% ethanol for 2 h immediately after harvesting. Roots were subsequently transferred to 20% (w/v) KOH, incubated for 3 days at 65 °C, and then rinsed five times with double-distilled H$_2$O with gentle agitation. After the final water rinse, roots were rinsed in phosphate-buffered saline for a minimum of 30 min, then stained with 0.2 mg/ml Wheat Germ Agglutinin (WGA)-Alexafluor 488 (Molecular Probes). Root length colonization measurements were performed using a gridline intersect method as described[64]. To count arbuscule size and abundance, three 500 μm - long root pieces were excised from a single root system and cut so that the appressorium defining the infection unit was centered in the root to ensure infection units were being assayed at the same infection stage. Three biological replicates from each condition were examined for a total of nine root pieces being assayed per condition. Infection units were imaged using a DM5500 Leica epifluorescence microscope using a 20X water immersion objective. Infection units were imaged at three focal planes to observe as many arbuscules as possible within a single infection unit. Arbuscule quantity and length was measured using the Measure tool in Fiji.

*vpy-4* roots transformed with *VPY$_{pro}$:cpVenus* (n = 15 root systems) and *VPY$_{pro}$:NLS-VPY-cpVenus* (n = 14 root systems) were stained and viewed on an Olympus SZX12 stereoscope. No arbuscules were present in any of these root systems. All infections were scored for passage through the epidermis (penetrating or non-penetrating hyphopodia). For infection units (IU) in the inner cortex, length was measured using an eye piece scale reticule at 10x magnification, zoom 63 (scale 100 units=1.5 mm). IUs longer than 100 units were scored as >100. Data were binned (1) 1-50 units (2) 50-100 unit (3) >100 units. For *NLS-VPY*, 50 IUs and for *cpVenus*, 44 IUs were measured.

*vpy-4* roots transformed with *VPY$_{pro}$:VPYcpVenus* and *VPY$_{pro}$:VPY-NES-cpVenus* were stained and viewed on an Olympus SZX12 stereoscope. All infections progressed to the inner cortex. Roots with several long IUs were mounted on slides in Mowiol 4-88 and squashed gently to break the root and release the intercellular hyphae and arbuscules to a single layer. Slides were analyzed on a DM5500B

fluorescence microscope (40 x objective). Using an eye piece reticule with scale and vertical intersect line, the scale was positioned at the leading edge of the infection unit. Starting at 75 μm behind the leading edge and then continuing at 150 μm intervals across the entire infection unit, intercellular hyphae and arbuscules were counted underneath the vertical intersect line (For *VPY-cpVenus*, 202 intersections and for *VPY-NES-cpVenus*, 221 intersections were scored from 7 transgenic roots for each line).

## Confocal microscopy

3–5 mm long root pieces with fluorescent signals were selected under an Olympus stereomicroscope, and longitudinally bisected using a double-edged razor and imaged as described previously[32,33]. Briefly, confocal images of excised root pieces in water were taken using a Leica TCS-SP5 confocal microscope with a 63X, water immersion objective (numerical aperture 1.2). cpVenus was excited using an Argon laser (514 nm), and emitted fluorescence collected was from 525 to 545 nm. mCherry was excited at 561 nm using a Diode-Pumped Solid State Laser emitted fluorescence collected from 600 to 660 nm. Differential interference contrast (DIC) images were taken simultaneously with fluorescence images. A minimum of two independent experiments, with a minimum of three root systems and ten root pieces were imaged per line.

For confocal imaging of WGA-Alexafluor 488 stained roots, Alexa Fluor 488 was excited with a 488 laser line and emitted fluorescence collected between 500-550 nm. Images were processed using the Fiji image software.

## Cloning

A detailed description of the methods used to clone the various genes and assemble the constructs used in these studies is described in the supplementary materials. For all constructs, the correct plasmid sequences were confirmed via restriction digest and sequencing. A list of plasmids used during for this study is listed in Table S2. The primers used to generate the plasmids are listed in Table S3.

## Promoter GUS analysis

Harvested roots were incubated in water on ice and then vacuum infiltrated for 30 min. Roots were then incubated on ice with 90% acetone for 15-30 minutes, and rinsed with phosphate buffer saline (PBS) three times for 10 min. Next, roots were incubated in GUS buffer (5 μM EDTA, 5 mM K-Ferrocyanide, 5 mM K-Ferricyanide, 0.5 mg/mL X-gluc dissolved in DMSO, in PBS) at 37 °C. Root staining was assessed 10 min, 30 min, 1 h, 2 h, and 6 h after staining and imaged using an Olympus SZX stereomicroscope.

## Co-immunoprecipitation Assays and Western Blots

For co-immunoprecipitation experiments, four-week-old *Nicotiana benthamiana* plants were transiently transformed with a 1:1 mixture of *Agrobacterium tumefaciens* strain GV3101 carrying the constructs of interest. Cultures were diluted to an $OD_{600}$ of 0.3 (for GFP, 0.05) in infiltration medium (10 mM MgCl2, 10 mM MES, 150 μM acetosyringone, pH 5.6). One full leaf from two different plants was infiltrated per assay using a needless syringe. Two days after infiltration, leaves were harvested, ground in liquid nitrogen, and 5 mL of extraction buffer (150 mM Tris-HCl, pH 7.5, 150 mM NaCl, 10% glycerol, 10 mM EDTA, 20 mM NaF, 10 mM DTT, 0.1% Tween-20, protease inhibitors (Sigma-Aldrich), and 1 mM PMSF) per 2 g of leaf tissue was added to tissue in the mortar and pestle. The buffer and tissue were homogenized, and when the mixture was thawed, samples were vortexed and centrifuged for 15 min at 3,200 g at 4 °C. The supernatant was filtered using a 0.45 μm sterile syringe filter (Corning), and 2 mL of extract was added to 20 μL GFP-Trap MA beads (Chromotek). Samples were incubated at 4 °C using an end-to-end rotator for 2 hrs. GFP-Trap

beads were magnetically separated from the protein extract and washed four times with wash buffer (10 mM Tris-HCl, pH 7.5, 150 mM NaCl, 0.5 mM EDTA, 0.1% Tween-20). After the fourth wash, 50 μL of 2X SDS sample buffer was added to the beads, and samples were heated at 95 °C for 5 min. To detect GFP-tagged proteins using Western blotting, we loaded 2 μL of the IP fraction, and 10 μL of the input. To detect HA-tagged proteins, we loaded 10 μL of the IP fraction, and 20 μL of the input.

For Western blotting, proteins were separated using a 1.5 mm 10% (w/v) SDS-PAGE gel run at 80 V for 2 h. Following SDS-PAGE, proteins were transferred onto a 0.45 μm Immobilon®FL PVDF membranes (Millipore) for 90 min at 250 mA using the BioRad minigel and blotting system. Membranes were blocked in 5% instant nonfat dry milk (w/v, Carnation milk, Nestlé) in PBS-T (1X PBS, 0.05% Tween-20) at room temperature for 1 h, and probed with the 1:1,000 primary antibody dilution (anti-GFP, Roche or anti-HA, Sigma-Aldrich) in 5% milk (w/v) in PBS-T at 4 °C overnight. Membranes were washed four times with PBST and incubated with peroxidase-conjugated anti-mouse secondary antibody (Promega) at a 1:10,000 dilution in either PBST (GFP primary) or PBST and 5% milk (HA primary) for 2 h. Following four rinses with PBST, antibody-conjugated proteins were detected using Immobilon Western Chemiluminescent HRP Substrate (Millipore) with film-based imaging (Bioblot).

## Recombinant protein production and affinity purification

To induce recombinant protein production, 100 mL of Rosetta BL DE1 *E. coli* cells were grown at 37 °C to $OD_{600}$ of 0.4–0.6, then treated with IPTG (final concentration 1 mM) and grown at 28 °C for 4 hrs. Cells were spun down at 4000 g for 10 min at 4 °C and frozen in liquid nitrogen. To extract proteins, cells were resuspended in 10 mL cell lysis buffer (50 mM Tris-Cl, pH 8.0, 1 M NaCl, 5% glycerol, 10 mM Imidazole, cOmplete™ Mini EDTA-free Protease Inhibitor Cocktail (Roche), lysozyme 1 mg/ml) and thawed on ice for 20 min. The cell slurry was sonicated, then spun down at 18,000 g for 30 min at 4 °C to remove cell wall debris. The cell lysate was used for subsequent affinity purification.

To purify proteins, cell extracts were pre-incubated with 1 mL Ni-NTA resin (Qiagen) for 1.5 h at 4 °C on a rotating shaker. Following incubation, the cell lysate-bead slurry was centrifuged for 1 min at 700 g. The supernatant was removed, and the resin was washed with 10 mL wash buffer (100 mM Tris-Cl pH 8.0, 1 M NaCl, 10% glycerol, 10 mM imidazole) twice. Then the resin was loaded onto a 10 mL Poly-Prep Chromatography column (BioRad). Lysate was allowed to flow through, and the resin was washed with another 5 mL wash buffer. Proteins were eluted in 4 ×0.5 mL elution buffer (50 mM Tris-Cl pH 8.0, 1 M NaCl, 10% glycerol, 250 mM imidazole). Protein concentration was determined using Bradford Quickstart Reagent (BioRad).

## in vitro kinase assays

For in vitro kinase assays, 1 μg of KINASE2 was incubated with 5 μg of protein substrate in kinase buffer (150 mM HEPES pH 8.0, 150 mM NaCl, 10 mM MgCl₂, 10 mM MnCl₂, 2 mM DTT, 20 μM ATP, 1 μCi of [$^{32}$P] γ-ATP (Perkin Elmer Co.)) for 1 h at room temperature. To detect kinase activity, 5 X SDS loading buffer was added to samples, and samples were run on a 1 mm 7.5% Precast SDS PAGE gel at 100 V for 1 h (BioRad). Gels were stained with Biosafe Coomassie stain (BioRad), dried, and exposed to a Phosphor screen overnight. Phosphor screens were analyzed using a Storm scanner.

## Amino acid alignments

To perform amino acid alignments, Clustal Omega was used (https://www.ebi.ac.uk/Tools/msa/clustalo/). Important structural features of ankyrin repeats were annotated using[69] as a reference point.

## Co-expression analysis

To generate candidates for targeted co-IP analysis, the Noble Foundation *Medicago truncatula* gene expression atlas was used (http://mtgea.noble.org). Two VPY probesets, Mtr.39050.1.S1_at and Mtr.42828.1.S1_at, were used as queries, and the ten most highly co-expressed transcripts based off of all samples from version 2 (64 experiments) and version 3 (274 experiments) of the atlas were obtained. Correlations were based on Pearson's correlation coefficient. Genes which appeared with both VPY probes and with both search queries were prioritized for co-immunoprecipitation experiments.

## Statistical analyses

Statistical analyses were performed in RStudio using the -lsmeans statistical package. The overlapping histogram was made in RStudio as described in[70].

## VPY protein structure predictions and visualization

To generate VPY protein structure predictions we used ColabFold, using default settings[71]. To verify the location of the ankyrin repeats we analyzed during this study, we used ChimeraX to view PDB files generated from ColabFold and search for amino acid sequences[72].

## Reporting summary

Further information on research design is available in the Nature Research Reporting Summary linked to this article.

## Data availability

The plant genetic materials and constructs used in this research will be made available on request (contact, mjh78@cornell.edu). Source data are provided with this paper.

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

## Acknowledgements

Financial support for this project was provided by the U.S. National Science Foundation grant no. IOS-1353367 and IOS-1733470 and IOS-2139351. P. L. Lindsay was supported by a United States Department of Agriculture–National Institute of Food and Agriculture Predoctoral Fellowship 2017-67011-26032, a Cornell Presidential Life Sciences Fellowship, and a Cornell Provost Diversity Fellowship. We thank Roslyn Noar and Jeon Hong for double backcrossing of the *vpy-4* TILLING line, Dierdra A. Daniels for generating the *M. truncatula* VPY-GFP transgenic line, Robyn Roberts and Fabian Giska for their guidance with in vitro kinase assays, Armando Bravo for helpful comments on the manuscript, and Lena Müller for assistance with R plots. The His-MBP destination vector was gift from the Gregory Martin lab (BTI).

## Author contributions

P.L.L. designed and conducted all experiments with the exception of those listed below, analyzed data, and wrote the manuscript. S.I. designed and conducted the KIN2 promoter GUS, KIN2-GFP subcellular localization, VPY$_{ANKΔ7-9}$ overexpression in *ipd3 ipd3l* experiments and co-immunoprecipitations experiments with Flag-tagged constructs. P.L.L. and S.I. designed and S.I. conducted NLS-VPY, VPY-NES localization and function experiments. N.P. performed the TGN and Golgi co-localization experiments. X.Z. performed the initial *vpy-4* point mutant characterization and preliminary ANK and VAP domain overexpression experiment. M.J.H. designed experiments, analyzed data and wrote the manuscript.

## Competing interests

The authors declare no competing interests.
