## [Peer Review File · Nature Communications]

Distinct ankyrin repeat subdomains control VAPYRIN locations and intracellular accommodation functions during arbuscular mycorrhizal symbiosisREVIEWER COMMENTS

Reviewer #1 (Remarks to the Author):

This paper is composed of two major parts. In the first part, subcellular localization of truncated VPY proteins was investigated with their function. The authors concluded that nuclear localization is essential for the VPY function. This finding is quite interesting but is supported by circumstantial evidence. For example the authors showed that VPY ANK Δ 5-cpVenus was unable to restore arbuscule development in vpy-4 mutant (Table 1 and Figure 3D), but can VPY ANK Δ 5 linked with a nuclear localization signal restore the arbuscule development in vpy-4? The possibility that VPY ANK Δ 5 lost its biological function outside the nucleus for example in punctate bodies has not been ruled out. More direct data is necessary to demonstrate the significance of nuclear localization.

In the second part, interactions of VPY with KIN2 and DMI3 were shown by immunoprecipitation. Protein interactions were examined only in the transient overexpression system of tobacco leaves. These interactions in roots are not clear, and biological and functional links between VPY and KIN2 and between VPY and DMI3 have not been shown. Although ipd3 ipd3l roots overexpressing VPY exhibited an interesting phenotype, the results shown in Fig 6D are very weak to demonstrate the link between VYP nuclear localization and interaction with DMI3.

Fig 5C: You mentioned that Medtr8g056020-GFP controls did not co-immunoprecipitate KINASE2-HA (L288). But the band is faintly visible in the control experiment. It is difficult to compare binding affinities or specificity when expression levels of bait proteins are different. This type of in vivo co-immunoprecipitation is tricky. Therefore, the statement in L 289 is misleading. Similarly, expression levels between KIN1 and KIN2 in Fig S5 and between VPY and VPY-related protein in Fig S6 are very different. Pull-down assay in vitro would be more suitable for comparing binding affinities.

Minor comments

L165: Please check the author's name (Baupaume). There is a spelling error.

Fig. 5: (F) and (G) in the figure legend are (C) and (D), respectively.

Reviewer #2 (Remarks to the Author):

The manuscript by Lindsay and colleagues reports on a detailed and convincing dissection of the structure and functions of so far rather mysterious protein, Vapyrin, which is expressed during AM colonization and required for the full development of the symbiosis.

The results represent a major advance in the field and - even if they do not clarify all aspects of this protein functions - narrow the focus on a few key roles for Vapyrin, providing rather suggestive work hypotheses to be developed in the near future.

Several aspects are novel, but maybe the most striking is ability of this protein to interact with both plasma membrane- and nuclear-localized kinases. This intriguing observation is supported by solid data and outlines Vapyrin as (one of) the missing link(s) between cytoplasm/membrane dynamics associated with fungal accommodation and the regulation of gene expression (and possibly endoreduplication, see below) in the nucleus.

Data presentation is very clear and supported by solid statistics, excellent imaging and a thorough investigation of Vapyrin single domain function based on a number of deletion constructs fused with fluorescent proteins.

Besides a few typos, I see a possible weakness in the discussion section, which is partly repetitive of concepts already emerging clearly from the results. My suggestion is to reduce these and use some of the saved space for additional considerations about putative Vapyrin functions. Among these, since I am signing this report, I dare suggesting a couple of points that are based on previous publications by

my own research group and may be useful to provide further arguments to the discussion. Still concerning the discussion, most of the final part is quite speculative. I anyway believe these speculations are fully justified and not necessary to explain the strong interest in defining in detail the cellular functions of Vapyrin to a non-expert reader.

Punctual comments and suggestions:

Abstract: the last sentence (Our data indicate...) sounds rather vague. I suggest to mention instead the proposed role of VPY as a functional connection between periarbuscular membrane domains and nucleoplasmic processes.

Line 81. I believe a semi column should be added before 'yet'.

Lines 97-98. The sentence sounds suspended: why is it important that a VPY-related gene is present in *Physcomitrella*?

Line 123 I believe this should read 'contribution to VPY functions'

Line 139 and later on in the discussion: VPY localization in the nuclei of neighboring cells appears extremely intriguing. Indeed, previous papers have highlighted the occurrence of cellular responses in these cells, and we have recently demonstrated that the same cells, in the same time frame, undergo several cycles of endoreduplication. Given VPY activity in the nucleus, it may be interesting to remark that nuclear events in early AM colonization are not limited to the regulation of gene expression, but also of cell cycle-related processes (Carotenuto et al., 2019 *New Phytol*). Furthermore, both processes are DMI3-dependent and VPY is being shown to interact with DMI3.

Line 144 and 371. The reference to EXO70I could be integrated with a citation of our own study (which also involved one of the authors of this manuscript) showing a comparable localization for another member of the Exocyst complex, EXO84b (Genre et al, 2012, *Plant and Cell Physiol*). In that paper we also showed crescent-shaped signals that closely recall those described in the present study.

Concerning the discussion and the intriguing role of VPY in both the periarbuscular membrane and the nucleus, have the authors investigated the relation between nuclear positioning and VPY localization? If -as it is becoming apparent - VPY shuttles between the two cell districts, this might take place when the nucleus is closest to the periarbuscular membrane, and we know that the nucleus keeps moving along the PPA axis and eventually positions in the middle of the arbuscule branches. My speculation (and a question to the authors) is that the puncta where VPY accumulates could be sites of VPY translocation from the membrane to the cytoskeleton and eventually to the nucleus.

Line 324. I think 'of' should be deleted from 'overexpressing of VPY ANKΔ7-9'

Line 327-328: as for the last sentence of the abstract, also here the statement appears weak. Elsewhere the authors explain more clearly the potential role as a negative regulator of DMI3. I think this could be mentioned here.

Lines 332-352 largely resume the results and could be reduced

Lines 363-364: 'fulfill late growth regulatory functions' can the authors explain more clearly what they mean here? This is a crucial point that deserves a few more words, in my opinion.

Line 389 and 393. The reference to LCOs alone as signaling molecules ruling AM interaction appears limiting. The common symbiotic signaling pathway is known to be activated at least as strongly by COs, which by the way are active in all AM hosts, unlike LCOs. This description may bias the global picture to a non-expert reader.

I hope my comments can help improve an already excellent manuscript.
Yours sincerely
Andrea Genre

Responses to Reviewers

We thank both reviewers for taking the time to review our manuscript and provide thoughtful comments. These have led to improvements.

REVIEWER COMMENTS

Reviewer #1 (Remarks to the Author):

This paper is composed of two major parts. In the first part, subcellular localization of truncated VPY proteins was investigated with their function. The authors concluded that nuclear localization is essential for the VPY function. This finding is quite interesting but is supported by circumstantial evidence. For example the authors showed that VPY ANK Δ 5-cpVenus was unable to restore arbuscule development in *vpy-4* mutant (Table 1 and Figure 3D), but can VPY ANK Δ 5 linked with a nuclear localization signal restore the arbuscule development in *vpy-4*?

The possibility that VPY ANK Δ 5 lost its biological function outside the nucleus for example in punctate bodies has not been ruled out. More direct data is necessary to demonstrate the significance of nuclear localization.

Response: We agree, we cannot rule out a loss of function either outside or inside the nucleus. Because of this, we decided to take a different approach to address the significance of VPYs nuclear location. We generated constructs which either force VPY to accumulate in the nucleus or prevent its accumulation in the nucleus. We then introduced these constructs into *vpy-4* and assessed the mycorrhizal phenotypes.

To direct VPY to the nucleus, we fused an NLS signal to VPY-cpVenus. We first confirmed the efficacy in *N. benthamiana* and then generated a construct driven VPY promoter and confirmed that it drives nuclear accumulation in Medicago roots. The NLS is efficient and we do not detect VPY-cpVenus in the cytoplasm. As expected, this construct does not complement *vpy-4* but it does alter the *vpy-4* mycorrhizal phenotype, actually increasing the severity. The NLS construct further increases the number of hyphopodia that are blocked at the epidermis and therefore decreases fungal entry into the root. It also decreases lateral spread of intercellular hyphae within the cortex. This NLS-VPY phenotype is consistent with the phenotype observed in *ipd3 ipd3l* overexpressing VPY.

To prevent VPY accumulation in the nucleus, we added a nuclear export signal (NES). To find one that was effective for effective for VPY, we generated fusions VPY-cpVenus fusions with three different NES motifs and screened them for efficacy in *N. benthamiana*. Just one motif was capable of preventing VPY accumulation in the nucleus. This construct was then cloned under the VPY promoter and we confirmed that functions likewise in *M. truncatula* roots. Introduction of the VPY-NES construct into *vpy-4* roots restores some arbuscule development but the phenotype differs from that of wildtype. Arbuscule numbers decrease and intercellular hyphae increase. This phenotype could arise from reduced penetration of cortical cells. The data are shown in Figure 4 and support a role for VPY in the nucleus and indicate that full VPY function requires VPY in both the nucleus and cytoplasm.

In the second part, interactions of VPY with KIN2 and DMI3 were shown by immunoprecipitation. Protein interactions were examined only in the transient overexpression system of tobacco leaves. These interactions in roots are not clear, and biological and functional links between VPY and KIN2 and between VPY and DMI3 have not been shown. Although ipd3 ipd31 roots overexpressing VPY exhibited an interesting phenotype, the results shown in Fig 6D are very weak to demonstrate the link between VYP nuclear localization and interaction with DMI3.

Response:

We took several approaches (several experiments) to evaluate VPY interactions with KIN2 and DMI3 via co-IP from Medicago roots but all attempts failed. Full details, with western blots, are provided at the end of this ‘response to reviewers’ document. Briefly, in a first experiment, we expressed the constructs from native or strong mycorrhiza-inducible promoters in a stable VPY promoter:VPY-GFP transgenic background to maintain the fully native situation. The roots were colonized, proteins were extracted and 4 extraction buffers were tested. In the input fractions, we could detect low levels of VPY-GFP, occasionally KIN2-Flag but not DMI3-FLAG (the constructs under constitutive promoters, work in *N. benthamiana*). We could immunoprecipitate VPY-GFP when the protein was extracted with RIPA but no co-immunoprecipitation of KIN2 or DMI3. In the less harsh buffer conditions (eg. HEPES, as used in *N. benthamiana*), we were unable to extract or immunoprecipitate VPY-GFP.

To increase expression, we generated new constructs expressing the proteins from strong constitutive promoters. We checked the constructs in *N. benthamiana* where they worked and we could co-immunoprecipitate VPY-GFP and KIN2-FLAG, as well as VPY-GFP and DMI3-FLAG. We then conducted roots transformations and grew 50 plants with transgenic root system for 4 weeks. Using extraction and co-IP conditions that were successful in *N. benthamiana*, we could immunoprecipitate a very low amount of VPY-GFP. We could detect DMI3-FLAG in the input when co-expressed with free GFP but not when co-expressed with VPY-GFP. Perhaps not surprisingly given the protein levels, co-immunoprecipitation failed. Even when these genes were expressed from strong, constitutive promoters, protein levels extractable from roots in buffers suitable for co-IP were very low. We conclude that this aspect prevents co-IP from roots. We have included the new *N. benthamiana* experiments with VPY-GFP and KIN2-FLAG, as well as VPY-GFP and DMI3-FLAG in the manuscript. They are useful as they demonstrate reciprocal co-IP and that co-IP was successful with a variety of tags.

(Please see ‘Appendix to response to reviewers’ for further details, including blots.)

Fig 5C: You mentioned that Medtr8g056020-GFP controls did not co-immunoprecipitate KINASE2-HA (L288). But the band is faintly visible in the control experiment. It is difficult to compare binding affinities or specificity when expression levels of bait proteins are different. This type of *in vivo* co-immunoprecipitation is tricky. Therefore, the statement in L 289 is misleading. Similarly, expression levels between KIN1 and KIN2 in Fig S5 and between VPY and VPY-related protein in Fig S6 are very different. Pull-down assay *in vitro* would be more suitable for comparing binding affinities.

Response: We have altered the sentence to point out the differences in input. Please note that very low levels of VPY-GFP are sufficient to co-IP KINASE2, while the ankyrin repeat control, even when abundant, results in a barely visible KIN2 band. Likewise, we have adjusted the description of KIN1 and KIN2 for accuracy.

Minor comments

L165: Please check the author's name (Baupaume). There is a spelling error.
Fig. 5: (F) and (G) in the figure legend are (C) and (D), respectively.

Response: These typos have been corrected, thanks.

Reviewer #2 (Remarks to the Author):

The manuscript by Lindsay and colleagues reports on a detailed and convincing dissection of the structure and functions of so far rather mysterious protein, Vapyrin, which is expressed during AM colonization and required for the full development of the symbiosis.

The results represent a major advance in the field and - even if they do not clarify all aspects of this protein functions - narrow the focus on a few key roles for Vapyrin, providing rather suggestive work hypotheses to be developed in the near future.

Several aspects are novel, but maybe the most striking is ability of this protein to interact with both plasma membrane- and nuclear-localized kinases. This intriguing observation is supported by solid data and outlines Vapyrin as (one of) the missing link(s) between cytoplasm/membrane dynamics associated with fungal accommodation and the regulation of gene expression (and possibly endoreduplication, see below) in the nucleus.

Data presentation is very clear and supported by solid statistics, excellent imaging and a thorough investigation of Vapyrin single domain function based on a number of deletion constructs fused with fluorescent proteins.

Besides a few typos, I see a possible weakness in the discussion section, which is partly repetitive of concepts already emerging clearly from the results. My suggestion is to reduce these and use some of the saved space for additional considerations about putative Vapyrin functions.

Among these, since I am signing this report, I dare suggesting a couple of points that are based on previous publications by my own research group and may be useful to provide further arguments to the discussion.

Still concerning the discussion, most of the final part is quite speculative. I anyway believe these speculations are fully justified and not necessary to explain the strong interest in defining in detail the cellular functions of Vapyrin to a non-expert reader.

Response: Thanks for the suggestions, we have modified the discussion to reduce repetition and incorporate additional speculations about function.

Punctual comments and suggestions:

Abstract: the last sentence (Our data indicate...) sounds rather vague. I suggest to mention instead

the proposed role of VPY as a functional connection between periarbuscular membrane domains and nucleoplasmic processes.

Line 81. I believe a semi column should be added before 'yet'.

Response: Abstract and typos have been adjusted.

Lines 97-98. The sentence sounds suspended: why is it important that a VPY-related gene is present in *Physcomitrella*?

Response: We have modified the sentence but the main reason to comment on *Physcomitrella* is that it provides an example of closely related gene from a non-host and provides some information about the developmental process that it is involved in .

Line 123 I believe this should read 'contribution to VPY functions'

Response: Missing word included, thanks.

Line 139 and later on in the discussion: VPY localization in the nuclei of neighboring cells appears extremely intriguing. Indeed, previous papers have highlighted the occurrence of cellular responses in these cells, and we have recently demonstrated that the same cells, in the same time frame, undergo several cycles of endoreduplication. Given VPY activity in the nucleus, it may be interesting to remark that nuclear events in early AM colonization are not limited to the regulation of gene expression, but also of cell cycle-related processes (Carotenuto et al., 2019 *New Phytol*). Furthermore, both processes are DMI3-dependent and VPY is being shown to interact with DMI3.

Response: We have included a reference to endoreduplication at the beginning of the discussion, but decided not to discuss VPY in relation this topic as endoreduplication data have not been included in the manuscript.

Line 144 and 371. The reference to EXO70I could be integrated with a citation of our own study (which also involved one of the authors of this manuscript) showing a comparable localization for another member of the Exocyst complex, EXO84b (Genre et al, 2012, *Plant and Cell Physiol*). In that paper we also showed crescent-shaped signals that closely recall those described in the present study.

Response: Thanks, we have included reference to EXO84 and the crescent-shaped signals, in the discussion session.

Concerning the discussion and the intriguing role of VPY in both the periarbuscular membrane and the nucleus, have the authors investigated the relation between nuclear positioning and VPY localization? If -as it is becoming apparent - VPY shuttles between the two cell districts, this

might take place when the nucleus is closest to the perifungal membrane, and we know that the nucleus keeps moving along the PPA axis and eventually positions in the middle of the arbuscule branches. My speculation (and a question to the authors) is that the puncta where VPY accumulates could be sites of VPY translocation from the membrane to the cytoskeleton and eventually to the nucleus.

Response. This is an intriguing suggestion and we look forward to future discussions on this topic. Currently, we do not have evidence for continuous movement of VPY between compartments. It is possible that this occurs, but it is also possible that there are two somewhat static pools of VPY (VPY could be transferred to the nucleus, possibly by another protein, and then remain there). Based on current observations, VPY accumulates in the nucleus first (in non-colonized neighbor cells). As we have not addressed this question, we decided not to speculate on shuttling mechanisms.

Line 324. I think ‘of’ should be deleted from ‘overexpressing of VPY ANK Δ 7-9’

Response: Typo corrected, thanks.

Line 327-328: as for the last sentence of the abstract, also here the statement appears weak. Elsewhere the authors explain more clearly the potential role as a negative regulator of DMI3. I think this could be mentioned here.

Response: We adjusted this sentence to include potential negative regulation of DMI3

Lines 332-352 largely resume the results and could be reduced

Lines 363-364: ‘fulfill late growth regulatory functions’ can the authors explain more clearly what they mean here? This is a crucial point that deserves a few more words, in my opinion.

Response: We have adjusted the discussion to remove redundancy. The ‘late growth regulatory functions’ are speculation, based on the observation that they are most abundant when the arbuscule has likely stopped growing.

Line 389 and 393. The reference to LCOs alone as signaling molecules ruling AM interaction appears limiting. The common symbiotic signaling pathway is known to be activated at least as strongly by COs, which by the way are active in all AM hosts, unlike LCOs. This description may bias the global picture to a non-expert reader.

Response: Yes, thanks, we have adjusted the text to include LCO and CO.

I hope my comments can help improve an already excellent manuscript.
Yours sincerely
Andrea Genre

Appendix to Response to Reviewers comments (further details of co-IP expts)

Detailed description of co-immunoprecipitation experiments from transgenic roots

To attempt co-immunoprecipitation in roots, we conducted two separate root transformation experiments. In the first experiment, we expressed either *KIN2pr:KIN2-FLAG*, *BCPpr:DMI3-FLAG*, or *BCPpr:GUS-FLAG* in *VPYpr:VPY-GFP* stable transgenic lines. In this experiment, we tried four different protein extraction buffers to determine which would be most effective at extracting proteins from *Medicago truncatula* roots (HEPES, RIPA, Tris-HCl, and sorbitol, see recipes below). For this experiment, 1 g of roots were ground in a mortar and pestle, and two volumes of extraction buffer were added to the mortar and pestle. Extraction buffer and tissue were homogenized in the mortar and pestle and transferred to 1.5 mL Eppendorf tubes. Samples were incubated in the cold for 30 min with end over end shaking, and then spun down for 10 min at 12,000 g. The supernatant was transferred to a new tube and spun for a second time for 5 min at 12,000 g. 50 μ L of the input fraction was saved. The supernatant was transferred to tubes with 20 μ L of GFP agarose trap beads and incubated for 1 hr on a rotator. Following incubation, we washed beads three times with the protein extraction buffer using Chromotek spin columns following the manufacturer's protocol. Proteins were eluted from GFP-trap beads by boiling beads in 80 μ L 2X SDS loading buffer. 20 μ L of the input and IP fraction were loaded on SDS-PAGE gels.

We had the most success pulling down VPY-GFP when proteins were extracted in RIPA buffer (**Fig 1A**), but we could not co-immunoprecipitate FLAG-tagged KIN2 or DMI3. RIPA is a harsh lysis buffer, so it's possible that protein-protein interactions were disturbed during the protein extraction process. When we tried gentler extraction buffers (Tris-HCl, HEPES) we were unable to detect VPY-GFP in either the input or immunoprecipitated fraction. We were unable to detect DMI3-FLAG in the input fraction regardless of the buffer we used. We know that the construct works, however, because when we expressed it in *N. benthamiana* leaves we could detect DMI3-FLAG (**Fig 1B**).

To attempt to boost protein levels, we next cloned double constructs containing either *AtUBQ10p:KIN2-FLAG* and *35S:VPY-GFP* or *AtUBQ10p:DMI3-FLAG* and *35S:VPY-GFP* to transform non-colonized roots. We first checked whether these constructs worked by expressing them in *N. benthamiana* leaves. Proteins could be detected and we could co-immunoprecipitate VPY-GFP and KIN2-FLAG, as well as VPY-GFP and DMI3-FLAG in two independent attempts (**Fig 2A-B**). We then conducted the root transformation. Fifty plants were grown in pots with surface for four weeks. For the protein extraction, we used HEPES extraction buffer, and ground 1 g of tissue in 3 mL extraction buffer. Similar to the first experiment, we were unable to detect DMI3-FLAG (**Figure 2C-D**). VPY-GFP was detected only very weakly after immunoprecipitation by GFP-trap magnetic agarose. KIN2-FLAG was also only weakly detected in one sample. We tried the extraction a second time with 7 g of tissue and 7 ml HEPES extraction buffer, and could detect DMI3-FLAG this time, but only in the tissue where free GFP was co-expressed with DMI3-FLAG (**Figure 2E**).

From these experiments, we suspect that VPY and DMI3 are likely post-translationally regulated in roots, because even when these genes were expressed from a constitutive promoter, protein levels were low, making it difficult to increase the protein abundance available for Co-IP experiments. While it would be ideal to demonstrate an interaction between the proteins *in vivo*, we think it may not be feasible at this time. We were able to successfully perform reciprocal Co-IP for both interactors, however, so this provides additional support for the interactions.

Figure 1. First co-immunoprecipitation attempt in *Medicago truncatula* roots.

Figure 2. Second co-immunoprecipitation attempt in *Medicago truncatula* roots.

Protein extraction buffers:**HEPES**

50 mM HEPES-KOH pH 7.5
150 mM NaCl
10 % glycerol
5 mM EDTA
10 mM NaF
1 mM Na₂MoO₄
10 mM DTT
1% Triton-X
Protease inhibitor cocktail
1 mM PMSF

RIPA

10 mM Tris-HCl pH 7.5
150 mM NaCl
0.1 % SDS
1 % Triton-X 100
1 % Sodium deoxycholate
5 mM EDTA
1 mM PMSF
20 mM NaF
1x protease inhibitor cocktail

Tris-HCl

150 mM Tris-HCl pH 7.5
150 mM NaCl
10 % glycerol
10 mM EDTA
20 mM NaF
10 mM DTT
1 % Triton-X
Protease inhibitor cocktail
1 mM PMSF

Sorbitol

230 mM sorbitol
50 mM HEPES
10 mM KCl
3 mM EGTA
pH 7.7
Protease inhibitor cocktail
1 mM PMSF
20 mM NaF

REVIEWERS' COMMENTS

Reviewer #1 (Remarks to the Author):

Regarding 3 major comments, I thought that the authors were carefully addressed with the additional experimental data and discussion.

Reviewer #2 (Remarks to the Author):

The Authors have convincingly addressed all my major remarks and I have no additional critic.